# Differentiable Random Partition Models

**Thomas M. Sutter**,* **Alain Ryser**\*, **Joram Liebeskind**, **Julia E. Vogt**
Department of Computer Science
ETH Zurich

## Abstract

Partitioning a set of elements into an unknown number of mutually exclusive subsets is essential in many machine learning problems. However, assigning elements, such as samples in a dataset or neurons in a network layer, to an unknown and discrete number of subsets is inherently non-differentiable, prohibiting end-to-end gradient-based optimization of parameters. We overcome this limitation by proposing a novel two-step method for inferring partitions, which allows its usage in variational inference tasks. This new approach enables reparameterized gradients with respect to the parameters of the new random partition model. Our method works by inferring the number of elements per subset and, second, by filling these subsets in a learned order. We highlight the versatility of our general-purpose approach on three different challenging experiments: variational clustering, inference of shared and independent generative factors under weak supervision, and multitask learning.

## 1 Introduction

Partitioning a set of elements into subsets is a classical mathematical problem that attracted much interest over the last few decades (Rota, 1964; Graham et al., 1989). A partition over a given set is a collection of non-overlapping subsets such that their union results in the original set. In machine learning (ML), partitioning a set of elements into different subsets is essential for many applications, such as clustering (Bishop and Svensen, 2004) or classification (De la Cruz-Mesía et al., 2007).

Random partition models (RPM, Hartigan, 1990) define a probability distribution over the space of partitions. RPMs can explicitly leverage the relationship between elements of a set, as they do not necessarily assume *i.i.d.* set elements. On the other hand, most existing RPMs are intractable for large datasets (MacQueen, 1967; Plackett, 1975; Pitman, 1996) and lack a reparameterization scheme, prohibiting their direct use in gradient-based optimization frameworks.

In this work, we propose the differentiable random partition model (DRPM), a fully-differentiable relaxation for RPMs that allows reparametrizable sampling. The DRPM follows a two-stage procedure: first, we model the number of elements per subset, and second, we learn an ordering of the elements with which we fill the elements into the subsets. The DRPM enables the integration of partition models into state-of-the-art ML frameworks and learning RPMs from data using stochastic optimization.

We evaluate our approach in three experiments, demonstrating the proposed DRPM's versatility and advantages. First, we apply the DRPM to a variational clustering task, highlighting how the reparametrizable sampling of partitions allows us to learn a novel kind of Variational Autoencoder (VAE, Kingma and Welling, 2014). By leveraging potential dependencies between samples in a dataset, DRPM-based clustering overcomes the simplified *i.i.d.* assumption of previous works, which used categorical priors (Jiang et al., 2016). In our second experiment, we demonstrate how to retrieve sets of shared and independent generative factors of paired images using the proposed DRPM. In

---

*Equal Contribution. Correspondence to {thomas.sutter,alain.ryser}@inf.ethz.ch

37th Conference on Neural Information Processing Systems (NeurIPS 2023).

$$\boldsymbol{n} = \begin{pmatrix} n_1 \\ n_2 \\ n_3 \\ n_4 \\ n_5 \end{pmatrix} \overset{e.\,g.}{=} \begin{pmatrix} 0 \\ 3 \\ 2 \\ 0 \\ 1 \end{pmatrix}$$

$$\pi = \begin{pmatrix} 0 & 1 & 0 & 0 & 0 & 0 \\ 1 & 0 & 0 & 0 & 0 & 0 \\ 0 & 0 & 0 & 0 & 1 & 0 \\ 0 & 0 & 1 & 0 & 0 & 0 \\ 0 & 0 & 0 & 0 & 0 & 1 \\ 0 & 0 & 0 & 1 & 0 & 0 \end{pmatrix}$$

$$\begin{pmatrix} 0 & 1 & 0 & 0 & 0 & 0 \\ 1 & 0 & 0 & 0 & 0 & 0 \\ 0 & 0 & 0 & 0 & 1 & 0 \end{pmatrix} = \bar{\pi}_2$$
$$\begin{pmatrix} 0 & 0 & 1 & 0 & 0 & 0 \\ 0 & 0 & 0 & 0 & 0 & 1 \end{pmatrix} = \bar{\pi}_3$$
$$\begin{pmatrix} 0 & 0 & 0 & 1 & 0 & 0 \end{pmatrix} = \bar{\pi}_5$$

$$\underbrace{\begin{pmatrix} 1 & 1 & 0 & 0 & 1 & 0 \\ 0 & 0 & 1 & 0 & 0 & 1 \\ 0 & 0 & 0 & 1 & 0 & 0 \end{pmatrix}}_{Y} = \begin{pmatrix} \boldsymbol{y}_2 \\ \boldsymbol{y}_3 \\ \boldsymbol{y}_5 \end{pmatrix}$$

Figure 1: Illustration of the proposed DRPM method. We first sample a permutation matrix $\pi$ and a set of subset sizes $\boldsymbol{n}$ separately in two stages. We then use $\boldsymbol{n}$ and $\pi$ to generate the assignment matrix $Y$, the matrix representation of a partition $\rho$.

contrast to previous works (Bouchacourt et al., 2018; Hosoya, 2018; Locatello et al., 2020), which rely on strong assumptions or heuristics, the DRPM enables end-to-end inference of generative factors. Finally, we perform multitask learning (MTL) by using the DRPM as a building block in a deterministic pipeline. We show how the DRPM learns to assign subsets of network neurons to specific tasks. The DRPM can infer the subset size per task based on its difficulty, overcoming the tedious work of finding optimal loss weights (Kurin et al., 2022; Xin et al., 2022).

To summarize, we introduce the DRPM, a novel differentiable and reparametrizable relaxation of RPMs. In extensive experiments, we demonstrate the versatility of the proposed method by applying the DRPM to clustering, inference of generative factors, and multitask learning.

## 2   Related Work

**Random Partition Models**   Previous works on RPMs include product partition models (Hartigan, 1990), species sampling models (Pitman, 1996), and model-based clustering approaches (Bishop and Svensen, 2004). Further, Lee and Sang (2022) investigate the balancedness of subset sizes of RPMs. They all require tedious manual adjustment, are non-differentiable, and are, therefore, unsuitable for modern ML pipelines. A fundamental RPM application is clustering, where the goal is to partition a given dataset into different subsets, the clusters. In contrast to many existing approaches (Yang et al., 2019; Sarfraz et al., 2019; Cai et al., 2022), we consider cluster assignments as random variables, allowing us to treat clustering from a variational perspective. Previous works in variational clustering (Jiang et al., 2016; Dilokthanakul et al., 2016; Manduchi et al., 2021) implicitly define RPMs to perform clustering. They compute partitions in a variational fashion by making *i.i.d.* assumptions about the samples in the dataset and imposing soft assignments of the clusters to data points during training. A problem related to set partitioning is the earth mover's distance problem (EMD, Monge, 1781; Rubner et al., 2000). However, EMD aims to assign a set's elements to different subsets based on a cost function and given subset sizes. Iterative solutions to the problem exist (Sinkhorn, 1964), and various methods have recently been proposed, e.g., for document ranking (Adams and Zemel, 2011) or permutation learning (Santa Cruz et al., 2017; Mena et al., 2018; Cuturi et al., 2019).

**Differentiable and Reparameterizable Discrete Distributions**   Following the proposition of the Gumbel-Softmax trick (GST, Jang et al., 2016; Maddison et al., 2017), interest in research around continuous relaxations for discrete distributions and non-differentiable algorithms rose. The GST enabled the reparameterization of categorical distributions and their integration into gradient-based optimization pipelines. Based on the same trick, Sutter et al. (2023) propose a differentiable formulation for the multivariate hypergeometric distribution. Multiple works on differentiable sorting procedures and permutation matrices have been proposed, e.g., Linderman et al. (2018); Prillo and Eisenschlos (2020); Petersen et al. (2021). Further, Grover et al. (2019) described the distribution over permutation matrices $p(\pi)$ for a permutation matrix $\pi$ using the Plackett-Luce distribution (PL, Luce, 1959; Plackett, 1975). Prillo and Eisenschlos (2020) proposed a computationally simpler variant of Grover et al. (2019). More examples of differentiable relaxations include the top-$k$ elements selection procedure (Xie and Ermon, 2019), blackbox combinatorial solvers (Pogančić et al., 2019), implicit likelihood estimations (Niepert et al., 2021), and $k$-subset sampling (Ahmed et al., 2022).

# 3 Preliminaries

**Set Partitions**  A partiton $\rho = (\mathcal{S}_1, \ldots, \mathcal{S}_K)$ of a set $[n] = \{1, \ldots, n\}$ with $n$ elements is a collection of $K$ subsets $\mathcal{S}_k \subseteq [n]$ where $K$ is *a priori* unknown (Mansour and Schork, 2016). For a partition $\rho$ to be valid, it must hold that

$$\mathcal{S}_1 \cup \cdots \cup \mathcal{S}_K = [n] \quad \text{and} \quad \forall k \neq l: \ \mathcal{S}_k \cap \mathcal{S}_l = \emptyset \tag{1}$$

In other words, every element $i \in [n]$ has to be assigned to precisely one subset $\mathcal{S}_k$. We denote the size of the $k$-th subset $\mathcal{S}_k$ as $n_k = |\mathcal{S}_k|$. Alternatively, we can describe a partition $\rho$ through an assignment matrix $Y = [\boldsymbol{y}_1, \ldots, \boldsymbol{y}_K]^T \in \{0, 1\}^{K \times n}$. Every row $\boldsymbol{y}_k \in \{0, 1\}^{1 \times n}$ is a multi-hot vector, where $\boldsymbol{y}_{ki} = 1$ assigns element $i$ to subset $\mathcal{S}_k$.

Within the scope of our work, we view a partition of a set of $n$ elements as a special case of the urn model. Here, the urn contains marbles with $n$ different colors, where each color corresponds to a subset in the partition. For each color, there are $n$ marbles corresponding to the potential elements of their color/subset. To derive a partition, we sample $n$ marbles without replacement from the urn and register the order in which we draw the colors. The color of the $i$-th marble then determines the subset to which element $i$ corresponds. Furthermore, we can constrain the partition to only $K$ subsets by taking an urn with only $K$ different colors.

**Probability distribution over subset sizes**  The multivariate non-central hypergeometric distribution (MVHG) describes sampling without replacement and allows to skew the importance of groups with an additional importance parameter $\boldsymbol{\omega}$ (Fisher, 1935; Wallenius, 1963; Chesson, 1976). The MVHG is an urn model and is described by the number of different groups $K \in \mathbb{N}$, the number of elements in the urn of every group $\boldsymbol{m} = [m_1, \ldots, m_K] \in \mathbb{N}^K$, the total number of elements in the urn $\sum_{k=1}^K m_k \in \mathbb{N}$, the number of samples to draw from the urn $n \in \mathbb{N}_0$, and the importance factor for every group $\boldsymbol{\omega} = [\omega_1, \ldots, \omega_K] \in \mathbb{R}_{0+}^K$ (Johnson, 1987). Then, the probability of sampling $\boldsymbol{n} = \{n_1, \ldots, n_K\}$, where $n_k$ describes the number of elements drawn from group $K$ is

$$p(\boldsymbol{n}; \boldsymbol{\omega}, \boldsymbol{m}) = \frac{1}{P_0} \prod_{k=1}^K \binom{m_k}{n_k} \omega_k^{n_k} \tag{2}$$

where $P_0$ is a normalization constant. Hence, the MVHG $p(\boldsymbol{n}; \boldsymbol{\omega}, \boldsymbol{m})$ allows us to model dependencies between different elements of a set since drawing one element from the urn influences the probability of drawing one of the remaining elements, creating interdependence between them. For the rest of the paper, we assume $\forall \, m_k \in \boldsymbol{m} : m_k = n$. We thus use the shorthand $p(\boldsymbol{n}; \boldsymbol{\omega})$ to denote the density of the MVHG. We refer to Appendix A.1 for more details.

**Probability distribution over Permutation Matrices**  Let $p(\pi)$ denote a distribution over permutation matrices $\pi \in \{0, 1\}^{n \times n}$. A permutation matrix $\pi$ is doubly stochastic (Marcus, 1960), meaning that its row and column vectors sum to 1. This property allows us to use $\pi$ to describe an order over a set of $n$ elements, where $\pi_{ij} = 1$ means that element $j$ is ranked at position $i$ in the imposed order. In this work, we assume $p(\pi)$ to be parameterized by scores $\boldsymbol{s} \in \mathbb{R}_+^n$, where each score $s_i$ corresponds to an element $i$. The order given by sorting $\boldsymbol{s}$ in decreasing order corresponds to the most likely permutation in $p(\pi; \boldsymbol{s})$. Sampling from $p(\pi; \boldsymbol{s})$ can be achieved by resampling the scores as $\tilde{s}_i = \beta \log s_i + g_i$ where $g_i \sim \text{Gumbel}(0, \beta)$ for fixed scale $\beta$, and sorting them in decreasing order. Hence, resampling scores $\boldsymbol{s}$ enables the resampling of permutation matrices $\pi$. The probability over orderings $p(\pi; \boldsymbol{s})$ is then given by (Thurstone, 1927; Luce, 1959; Plackett, 1975; Yellott, 1977)

$$p(\pi; \boldsymbol{s}) = p((\pi\tilde{\boldsymbol{s}})_1 \geq \cdots \geq (\pi\tilde{\boldsymbol{s}})_n) = \frac{(\pi\boldsymbol{s})_1}{Z} \frac{(\pi\boldsymbol{s})_2}{Z - (\pi\boldsymbol{s})_1} \cdots \frac{(\pi\boldsymbol{s})_n}{Z - \sum_{j=1}^{n-1}(\pi\boldsymbol{s})_j} \tag{3}$$

where $\pi$ is a permutation matrix and $Z = \sum_{i=1}^n s_i$. The resulting distribution is a Plackett-Luce (PL) distribution (Luce, 1959; Plackett, 1975) if and only if the scores $\boldsymbol{s}$ are perturbed with noise drawn from Gumbel distributions with identical scales (Yellott, 1977). For more details, we refer to Appendix A.2).

# 4 A two-stage Approach to Random Partition Models

We propose the DRPM $p(Y; \boldsymbol{\omega}, \boldsymbol{s})$, a differentiable and reparameterizable two-stage Random Partition Model (RPM). The proposed formulation separately infers the number of elements $i$ per subset

$\boldsymbol{n} \in \mathbb{N}_0^K$, where $\sum_{k=1}^K n_k = n$, and the assignment of elements to subsets $\mathcal{S}_k$ by inducing an order on the $n$ elements and filling $\mathcal{S}_1, ..., \mathcal{S}_K$ sequentially in this order. To model the order of the elements, we use a permutation matrix $\pi = [\boldsymbol{\pi}_1, \ldots, \boldsymbol{\pi}_n]^T \in \{0, 1\}^{n \times n}$, from which we infer $Y$ by sequentially summing up rows according to $\boldsymbol{n}$. Note that the doubly-stochastic property of all permutation matrices $\pi$ ensures that the columns of $Y$ remain one-hot vectors, assigning every element $i$ to precisely one of the $K$ subsets. At the same time, the $k$-th row of $Y$ corresponds to an $n_k$-hot vector $\boldsymbol{y}_k$ and therefore serves as a subset selection vector, i.e.

$$\boldsymbol{y}_k = \sum_{i=\nu_k+1}^{\nu_k+n_k} \boldsymbol{\pi}_i, \quad \text{where} \quad \nu_k = \sum_{\iota=1}^{k-1} n_\iota \tag{4}$$

such that $Y = [\boldsymbol{y}_1, \ldots, \boldsymbol{y}_K]^T$. Additionally, Figure 1 provides an illustrative example. Note that $K$ defines the maximum number of possible subsets, and not the effective number of non-empty subsets, because we allow $\mathcal{S}_k$ to be the empty set $\emptyset$ (Mansour and Schork, 2016). We base the following Proposition 4.1 on the MVHG distribution $p(\boldsymbol{n}; \boldsymbol{\omega})$ for the subset sizes $\boldsymbol{n}$ and the PL distribution $p(\pi; \boldsymbol{s})$ for assigning the elements to subsets. However, the proposed two-stage approach to RPMs is not restricted to these two classes of probability distributions.

**Proposition 4.1** (Two-stage Random Partition Model). *Given a probability distribution over subset sizes $p(\boldsymbol{n}; \boldsymbol{\omega})$ with $\boldsymbol{n} \in \mathbb{N}_0^K$ and distribution parameters $\boldsymbol{\omega} \in \mathbb{R}_+^K$ and a PL probability distribution over random orderings $p(\pi; \boldsymbol{s})$ with $\pi \in \{0, 1\}^{n \times n}$ and distribution parameters $\boldsymbol{s} \in \mathbb{R}_+^n$, the probability mass function $p(Y; \boldsymbol{\omega}, \boldsymbol{s})$ of the two-stage RPM is given by*

$$p(Y; \boldsymbol{\omega}, \boldsymbol{s}) = p(\boldsymbol{y}_1, \ldots, \boldsymbol{y}_K; \boldsymbol{\omega}, \boldsymbol{s}) = p(\boldsymbol{n}; \boldsymbol{\omega}) \sum_{\pi \in \Pi_Y} p(\pi; \boldsymbol{s}) \tag{5}$$

*where $\Pi_Y = \{\pi : \boldsymbol{y}_k = \sum_{i=\nu_k+1}^{\nu_k+n_k} \boldsymbol{\pi}_i, k = 1, \ldots, K\}$, and $\boldsymbol{y}_k$ and $\nu_k$ as in Equation (4).*

In the following, we outline the proof of Proposition 4.1 and refer to Appendix B for a formal derivation. We calculate $p(Y; \boldsymbol{\omega}, \boldsymbol{s})$ as a probability of subsets $p(\boldsymbol{y}_1, \ldots, \boldsymbol{y}_K; \boldsymbol{\omega}, \boldsymbol{s})$, which we compute sequentially over subsets, i.e.

$$p(\boldsymbol{y}_1, \ldots, \boldsymbol{y}_K; \boldsymbol{\omega}, \boldsymbol{s}) = p(\boldsymbol{y}_1; \boldsymbol{\omega}, \boldsymbol{s}) \cdots p(\boldsymbol{y}_K \mid \boldsymbol{y}_{<K}; \boldsymbol{\omega}, \boldsymbol{s}), \tag{6}$$

where $\boldsymbol{y}_{<k} = [\boldsymbol{y}_1, \ldots, \boldsymbol{y}_{k-1}]$ and

$$p(\boldsymbol{y}_k \mid \boldsymbol{y}_{<k}; \boldsymbol{\omega}, \boldsymbol{s}) = p(n_k \mid n_{<k}; \boldsymbol{\omega}) \sum_{\bar{\pi} \in \Pi_{\boldsymbol{y}_k}} p(\bar{\pi} \mid n_k, \boldsymbol{y}_{<k}; \boldsymbol{s}), \tag{7}$$

where $\Pi_{\boldsymbol{y}_k}$ in Equation (7) is the set of all subset permutations of elements $i \in \mathcal{S}_k$. A subset permutation matrix $\bar{\pi}$ represents an ordering over only $n_k$ out of the total $n$ elements. The probability $p(\boldsymbol{y}_k \mid \boldsymbol{y}_{<k}; \boldsymbol{\omega}, \boldsymbol{s})$ describes the probability of a subset of a given size $n_k$ by marginalizing over the probabilities of all subset permutations $p(\bar{\pi} \mid n_k, \boldsymbol{y}_{<k}; \boldsymbol{s})$. Hence, the sum over all $p(\bar{\pi} \mid n_k, \boldsymbol{y}_{<k}; \boldsymbol{s})$ makes $p(\boldsymbol{y}_k \mid \boldsymbol{y}_{<k}; \boldsymbol{\omega}, \boldsymbol{s})$ invariant to the ordering of elements $i \in \mathcal{S}_k$ (Xie and Ermon, 2019). Note that in a slight abuse of notation, we use $p(\bar{\pi} \mid n_k, \boldsymbol{y}_{<k}; \boldsymbol{\omega}, \boldsymbol{s})$ as the probability of a subset permutation $\bar{\pi}$ given that there are $n_k$ elements in $\mathcal{S}_k$ and thus $\bar{\pi} \in \{0, 1\}^{n_k \times n}$.

The probability of a subset permutation matrix $p(\bar{\pi} \mid n_k, \boldsymbol{y}_{<k}; \boldsymbol{s})$ describes the probability of drawing the elements $i \in \mathcal{S}_k$ in the order defined by the subset permutation matrix $\bar{\pi}$ given that the elements in $\mathcal{S}_{<k}$ are already determined. Hence, we condition on the subsets $\boldsymbol{y}_{<k}$. This property follows from Luce's choice axiom (LCA, Luce, 1959). Additionally, we condition on $n_k$, the size of the subset $\mathcal{S}_k$. The probability of a subset permutation is given by

$$p(\bar{\pi} \mid n_k, \boldsymbol{y}_{<k}; \boldsymbol{s}) = \prod_{i=1}^{n_k} \frac{(\bar{\pi}\boldsymbol{s})_i}{Z_k - \sum_{j=1}^{i-1} (\bar{\pi}\boldsymbol{s})_j} \tag{8}$$

In contrast to the distribution over permutations matrices $p(\pi; \boldsymbol{s})$ in Equation (3), we compute the product over $n_k$ terms and have a different normalization constant $Z_k$, which is the sum over the scores $s_i$ of all elements $i \in \mathcal{S}_k$. Although we induce an ordering over all elements $i$ by using a permutation matrix $\pi$, the probability $p(\boldsymbol{y}_k \mid \boldsymbol{y}_{<k}; \boldsymbol{\omega}, \boldsymbol{s})$ is invariant to intra-subset orderings of elements $i \in \mathcal{S}_k$. Finally, we arrive at Equation (5) by substituting Equation (7) into Equation (6),

and applying the definition of the conditional probability $p(\boldsymbol{n}; \boldsymbol{\omega}) = \prod_{k=1}^{K} p(n_k \mid n_{<k}; \boldsymbol{\omega})$ and by reshuffling indices $\sum_{\pi \in \Pi_Y} p(\pi; \boldsymbol{s}) = \prod_{k=1}^{K} \sum_{\bar{\pi} \in \Pi_{\boldsymbol{y}_k}} p(\bar{\pi} \mid n_k, \boldsymbol{y}_{<k}; \boldsymbol{s})$.

Note that in contrast to previous RPMs, which often need exponentially many distribution parameters (Plackett, 1975), the proposed two-stage approach to RPMs only needs $(n + K)$ parameters to create an RPM for $n$ elements: the score parameters $\boldsymbol{s} \in \mathbb{R}_+^n$ and the group importance parameters $\boldsymbol{\omega} \in \mathbb{R}_+^K$.

Finally, to sample from the two-stage RPM of Proposition 4.1 we apply the following procedure: First sample $\pi \sim p(\pi; \boldsymbol{s})$ and $\boldsymbol{n} \sim p(\boldsymbol{n}; \boldsymbol{\omega})$. From $\pi$ and $\boldsymbol{n}$, compute partition $Y$ by summing the rows of $\pi$ according to $\boldsymbol{n}$ as described in Equation (4) and illustrated in Figure 1.

### 4.1 Approximating the Probability Mass Function

The number of permutations per subset $|\Pi_{\boldsymbol{y}_k}|$ scales factorially with the subset size $n_k$, i.e. $|\Pi_{\boldsymbol{y}_k}| = n_k!$. Consequently, the number of valid permutation matrices $|\Pi_Y|$ is given as a function of $\boldsymbol{n}$, i.e.

$$|\Pi_Y| = \prod_{k=1}^{K} |\Pi_{\boldsymbol{y}_k}| = \prod_{k=1}^{K} n_k! \tag{9}$$

Although Proposition 4.1 describes a well-defined distribution for $p(Y; \boldsymbol{\omega}, \boldsymbol{s})$, it is in general computationally intractable due to Equation (9). In practice, we thus approximate $p(Y; \boldsymbol{\omega}, \boldsymbol{s})$ using the following Lemma.

**Lemma 4.2.** $p(Y; \boldsymbol{\omega}, \boldsymbol{s})$ *can be upper and lower bounded as follows*

$$\forall \pi \in \Pi_Y : \ p(\boldsymbol{n}; \boldsymbol{\omega}) p(\pi; \boldsymbol{s}) \ \leq \ p(Y; \boldsymbol{\omega}, \boldsymbol{s}) \ \leq \ |\Pi_Y| p(\boldsymbol{n}; \boldsymbol{\omega}) \max_{\tilde{\pi}} p(\tilde{\pi}; \boldsymbol{s}) \tag{10}$$

We provide the proof in Appendix B. Note that from Equation (3) we see that $\max_{\tilde{\pi}} p(\tilde{\pi}; \boldsymbol{s}) = p(\pi_{\boldsymbol{s}}; \boldsymbol{s})$, where $\pi_{\boldsymbol{s}}$ is the permutation that results from sorting the unperturbed scores $\boldsymbol{s}$.

### 4.2 The Differentiable Random Partition Model

To incorporate our two-stage RPM into gradient-based optimization frameworks, we require that efficient computation of gradients is possible for every step of the method. The following Lemma guarantees differentiability, allowing us to train deep neural networks with our method in an end-to-end fashion:

**Lemma 4.3** (DRPM). *A two-stage RPM is differentiable and reparameterizable if the distribution over subset sizes $p(\boldsymbol{n}; \boldsymbol{\omega})$ and the distribution over orderings $p(\pi; \boldsymbol{s})$ are differentiable and reparameterizable.*

We provide the proof in Appendix B. Note that Lemma 4.3 enables us to learn variational posterior approximations and priors using Stochastic Gradient Variational Bayes (SGVB, Kingma and Welling, 2014). In our experiments, we apply Lemma 4.3 using the recently proposed differentiable formulations of the MVHG (Sutter et al., 2023) and the PL distribution (Grover et al., 2019), though other choices would also be valid.

## 5 Experiments

We demonstrate the versatility and effectiveness of the proposed DRPM in three different experiments. First, we propose a novel generative clustering method based on the DRPM, which we compare against state-of-the-art variational clustering methods and demonstrate its conditional generation capabilities. Then, we demonstrate how the DRPM can infer shared and independent generative factors under weak supervision. Finally, we apply the DRPM to multitask learning (MTL), where the DRPM enables an adaptive neural network architecture that partitions layers based on task difficulty[2].

---

[2]We provide the code under `https://github.com/thomassutter/drpm`

Table 1: We compare the clustering performance of the DRPM-VC on test sets of MNIST and FMNIST between Gaussian Mixture Models (GMM), GMM in latent space (Latent GMM), and Variational Deep Embedding (VaDE). We measure performance in terms of the Normalized Mutual Information (NMI), Adjusted Rand Index (ARI), and cluster accuracy (ACC) over five seeds and put the best model in bold.

|  | MNIST | | | FMNIST | | |
| --- | --- | --- | --- | --- | --- | --- |
|  | NMI | ARI | ACC | NMI | ARI | ACC |
| GMM | $0.32_{\pm0.01}$ | $0.22_{\pm0.02}$ | $0.41_{\pm0.01}$ | $0.49_{\pm0.01}$ | $0.33_{\pm0.00}$ | $0.44_{\pm0.01}$ |
| LATENT GMM | $0.86_{\pm0.02}$ | $0.83_{\pm0.06}$ | $0.88_{\pm0.07}$ | $0.60_{\pm0.00}$ | $0.47_{\pm0.01}$ | $0.62_{\pm0.01}$ |
| VADE | $0.84_{\pm0.01}$ | $0.76_{\pm0.05}$ | $0.82_{\pm0.04}$ | $0.56_{\pm0.02}$ | $0.40_{\pm0.04}$ | $0.56_{\pm0.03}$ |
| DRPM-VC | $\mathbf{0.89}_{\pm0.01}$ | $\mathbf{0.88}_{\pm0.03}$ | $\mathbf{0.94}_{\pm0.02}$ | $\mathbf{0.64}_{\pm0.00}$ | $\mathbf{0.51}_{\pm0.01}$ | $\mathbf{0.65}_{\pm0.00}$ |

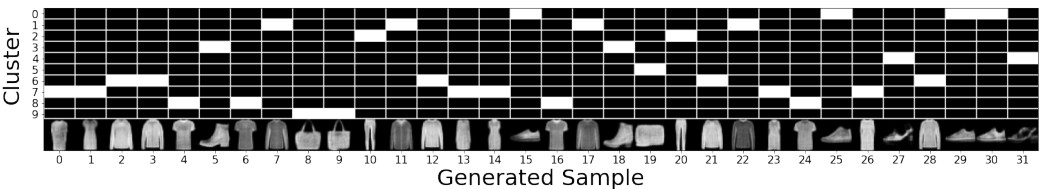

Figure 2: A sample drawn from a DRPM-VC model trained on FMNIST. On top is the sampled partition with the cluster assignments, and on the bottom are generated images corresponding to the sampled assignment matrix. The DRPM-VC learns consistent clusters for different pieces of clothing and can generate new samples of each cluster with great variability.

## 5.1 Variational Clustering with Random Partition Models

In our first experiment, we introduce a new version of a Variational Autoencoder (VAE, Kingma and Welling, 2014), the DRPM Variational Clustering (DRPM-VC) model. The DRPM-VC enables clustering and unsupervised conditional generation in a variational fashion. To that end, we assume that each sample $\boldsymbol{x}$ of a dataset $X$ is generated by a latent vector $\boldsymbol{z} \in \mathbb{R}^l$, where $l \in \mathbb{N}$ is the latent space size. Traditional VAEs would then assume that all latent vectors $\boldsymbol{z}$ are generated by a single Gaussian prior distribution $\mathcal{N}(\boldsymbol{0}, \mathbb{I}_l)$. Instead, we assume every $\boldsymbol{z}$ to be sampled from one of $K$ different latent Gaussian distributions $\mathcal{N}(\boldsymbol{\mu}_k, \mathrm{diag}(\boldsymbol{\sigma}_k)), k = 1, \ldots, K$, with $\boldsymbol{\mu}_k, \boldsymbol{\sigma}_k \in \mathbb{R}^l$. Further, note that similar to an urn model (Section 3), if we draw a batch from a given finite dataset with samples from different clusters, the cluster assignments within that batch are not entirely independent. Since there is only a finite number of samples per cluster, drawing a sample from a specific cluster decreases the chance of drawing a sample from that cluster again, and the distribution of the number of samples drawn per cluster will follow an MVHG distribution. Previous work on variational clustering proposes to model the cluster assignment $\boldsymbol{y} \in \{0,1\}^K$ of each sample $\boldsymbol{x}$ through independent categorical distributions (Jiang et al., 2016), which might thus be over-restrictive and not correctly reflect reality. Instead, we propose explicitly modeling the dependency between the $\boldsymbol{y}$ of different samples by assuming they are drawn from an RPM. Hence, the generative process leading to $X$ can be summarized as follows: First, the cluster assignments are represented as a partition matrix $Y$ and sampled from our DRPM, i.e., $Y \sim p(Y; \boldsymbol{\omega}, \boldsymbol{s})$. Given an assignment $\boldsymbol{y}$ from $Y$, we can sample the respective latent variable $\boldsymbol{z}$, where $\boldsymbol{z} \sim \mathcal{N}(\boldsymbol{\mu_y}, \mathrm{diag}(\boldsymbol{\sigma_y})), \boldsymbol{z} \in \mathbb{R}^l$. Note that we use the notational shorthand $\boldsymbol{\mu_y} := \boldsymbol{\mu}_{\arg\max(\boldsymbol{y})}$. Like in vanilla VAEs, we infer $\boldsymbol{x}$ by independently passing the corresponding $\boldsymbol{z}$ through a decoder model. Assuming this generative process, we derive the following evidence lower bound (ELBO) for $p(X)$:

$$\mathcal{L}_{ELBO} = \sum_{\boldsymbol{x} \in X} \mathbb{E}_{q(\boldsymbol{z}|\boldsymbol{x})}\left[\log p(\boldsymbol{x}|\boldsymbol{z})\right] - \sum_{\boldsymbol{x} \in X} \mathbb{E}_{q(Y|X)}\left[KL[q(\boldsymbol{z}|\boldsymbol{x})||p(\boldsymbol{z}|Y)]\right] - KL[q(Y|X)||p(Y)]$$

Note that computing $KL[q(Y|X)||p(Y)]$ directly is computationally intractable, and we need to upper bound it according to Lemma 4.2. For an illustration of the generative assumptions and more details on the ELBO, we refer to Appendix C.2.

Table 2: Partitioning of Generative Factors. We evaluate the learned latent representations of the four methods (Label-VAE, Ada-VAE, HG-VAE, DRPM-VAE) with respect to the shared (S) and independent (I) generative factors. We do this by fitting linear classifiers on the shared and independent dimensions of the representation, predicting the respective generative factors. We report the results in adjusted balanced accuracy (Sutter et al., 2023) across five seeds.

| | $n_s = 0$ | $n_s = 1$ | | $n_s = 3$ | | $n_s = 5$ | |
| | I | S | I | S | I | S | I |
|---|---|---|---|---|---|---|---|
| LABEL | $0.14_{\pm 0.01}$ | $0.19_{\pm 0.03}$ | $0.16_{\pm 0.01}$ | $0.10_{\pm 0.00}$ | $0.23_{\pm 0.01}$ | $0.34_{\pm 0.00}$ | $0.00_{\pm 0.00}$ |
| ADA | $0.12_{\pm 0.01}$ | $0.19_{\pm 0.01}$ | $0.15_{\pm 0.01}$ | $0.10_{\pm 0.03}$ | $0.22_{\pm 0.02}$ | $0.33_{\pm 0.03}$ | $0.00_{\pm 0.00}$ |
| HG | $0.18_{\pm 0.01}$ | $0.22_{\pm 0.05}$ | $0.19_{\pm 0.01}$ | $0.08_{\pm 0.02}$ | $0.28_{\pm 0.01}$ | $0.28_{\pm 0.01}$ | $0.01_{\pm 0.00}$ |
| DRPM | $\mathbf{0.26}_{\pm 0.02}$ | $\mathbf{0.39}_{\pm 0.07}$ | $\mathbf{0.2}_{\pm 0.01}$ | $\mathbf{0.15}_{\pm 0.01}$ | $\mathbf{0.29}_{\pm 0.02}$ | $\mathbf{0.42}_{\pm 0.03}$ | $0.01_{\pm 0.00}$ |

To assess the clustering performance, we train our model on two different datasets, namely MNIST (LeCun et al., 1998) and Fashion-MNIST (FMNIST, Xiao et al., 2017), and compare it to three baselines. Two of the baselines are based on a Gaussian Mixture Model, where one is directly trained on the original data space (GMM), whereas the other takes the embeddings from a pretrained encoder as input (Latent GMM). The third baseline is Variational Deep Embedding (VaDE, Jiang et al., 2016), which is similar to the DRPM-VC but assumes *i.i.d.* categorical cluster assignments. For all methods except GMM, we use the weights of a pretrained encoder to initialize the models and priors at the start of training. We present the results of these experiments in Table 1. As can be seen, we outperform all baselines, indicating that modeling the inherent dependencies implied by finite datasets benefits the performance of variational clustering. While achieving decent clustering performance, another benefit of variational clustering methods is that their reconstruction-based nature intrinsically allows unsupervised conditional generation. In Figure 2, we present the result of sampling a partition and the corresponding generations from the respective clusters after training the DRPM-VC on FMNIST. The model produces coherent generations despite not having access to labels, allowing us to investigate the structures learned by the model more closely. We refer to Appendix C.2 for more illustrations of the learned clusters, details on the training procedure, and ablation studies.

## 5.2 Variational Partitioning of Generative Factors

Data modalities not collected as *i.i.d.* samples, such as consecutive frames in a video, provide a weak-supervision signal for generative models and representation learning (Sutter et al., 2023). Here, on top of learning meaningful representations of the data samples, we are also interested in discovering the relationship between coupled samples. If we assume that the data is generated from underlying generative factors, weak supervision comes from the fact that we know that certain factors are shared between coupled pairs while others are independent. The supervision is weak because we neither know the underlying generative factors nor the number of shared and independent factors. In such a setting, we can use the DRPM to learn a partition of the generative factors and assign them to be either shared or independent.

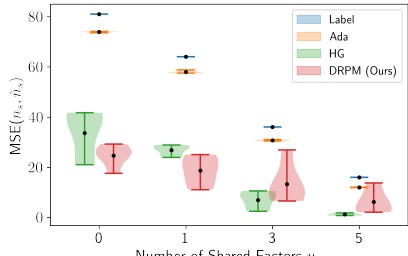

Figure 3: The mean squared errors between the estimated number of shared factors $\hat{n}_s$ and the true number of shared factors $n_s$ across five seeds for the Label-VAE, Ada-VAE, HG-VAE, and DRPM-VAE.

In this experiment, we use paired frames $\boldsymbol{X} = [\boldsymbol{x}_1, \boldsymbol{x}_2]$ from the *mpi3d* dataset (Gondal et al., 2019). Every pair of frames shares a subset of its seven generative factors. We introduce the DRPM-VAE, which models the division of the latent space into shared and independent latent factors as RPM. We add a posterior approximation $q(Y \mid \boldsymbol{X})$ and additionally a prior distribution of the form $p(Y)$. The model maximizes the following ELBO on the

marginal log-likelihood of images through a VAE (Kingma and Welling, 2014):

$$\mathcal{L}_{ELBO} = \sum_{j=1}^{2} \mathbb{E}_{q(\boldsymbol{z}_s, \boldsymbol{z}_j, Y | \boldsymbol{X})} \left[ \log p(\boldsymbol{x}_j \mid \boldsymbol{z}_s, \boldsymbol{z}_j) \right] \tag{11}$$
$$- \mathbb{E}_{q(Y|\boldsymbol{X})} \left[ KL \left[ q(\boldsymbol{z}_s, \boldsymbol{z}_1, \boldsymbol{z}_2 \mid Y, \boldsymbol{X}) || p(\boldsymbol{z}_s, \boldsymbol{z}_1, \boldsymbol{z}_2) \right] \right] - KL \left[ q(Y \mid \boldsymbol{X}) || p(Y) \right]$$

Similar to the ELBO for variational clustering in Section 5.1, computing $KL\left[q(Y \mid \boldsymbol{X}) || p(Y)\right]$ directly is intractable, and we need to bound it according to Lemma 4.2.

We compare the proposed DRPM-VAE to three methods, which only differ in how they infer shared and latent dimensions. While the Label-VAE (Bouchacourt et al., 2018; Hosoya, 2018) assumes that the number of independent factors is known, the Ada-VAE (Locatello et al., 2020) relies on a heuristic-based approach to infer shared and independent latent factors. Like in Locatello et al. (2020) and Sutter et al. (2023), we assume a single known factor for Label-VAE in all experiments. HG-VAE (Sutter et al., 2023) also relies on the MVHG to model the number of shared and independent factors. Unlike the proposed DRPM-VAE approach, HG-VAE must rely on a heuristic to assign latent dimensions to shared factors, as the MVHG only allows to model the number of shared and independent factors but not their position in the latent vector. We use the code from Locatello et al. (2020) and follow the evaluation in Sutter et al. (2023). We refer to Appendix C.3 for details on the ELBO, the setup of the experiment, the implementation, and an illustration of the generative assumptions.

We evaluate all methods according to their ability to estimate the number of shared generative factors (Figure 3) and how well they partition the latent representations into shared and independent factors (Table 2). Because we have access to the data-generating process, we can control the number of shared $n_s$ and independent $n_i$ factors. We compare the methods on four different datasets with $n_s \in \{0, 1, 3, 5\}$. In Figure 3, we demonstrate that the DRPM-VAE accurately estimates the true number of shared generative factors. It matches the performance of HG-VAE and outperforms the other two baselines, which consistently overestimate the true number of shared factors. In Table 2, we see a considerable performance improvement compared to previous work when assessing the learned latent representations. We attribute this to our ability to not only estimate the subset sizes of latent and shared factors like HG-VAE but also learn to assign specific latent dimensions to the corresponding shared or independent representations. Thus, the DRPM-VAE dynamically learns more meaningful representations and can better separate and infer the shared and independent subspaces for all dataset versions.

The DRPM-VAE provides empirical evidence of how RPMs can leverage weak supervision signals by learning to maximize the data likelihood while also inferring representations that capture the relationship between coupled data samples. Additionally, we can explicitly model the data-generating process in a theoretically grounded fashion instead of relying on heuristics.

## 5.3 Multitask Learning

Many ML applications aim to solve specific tasks, where we optimize for a single objective while ignoring potentially helpful information from related tasks. Multitask learning (MTL) aims to improve the generalization across all tasks, including the original one, by sharing representations between related tasks (Caruana, 1993; Caruana and de Sa, 1996) Recent works (Kurin et al., 2022; Xin et al., 2022) show that it is difficult to outperform a convex combination of task losses if the task losses are appropriately scaled. I.e., in case of equal difficulty of the two tasks, a classifier with equal weighting of the two classification losses serves as an upper bound in terms of performance. However, finding suitable task weights is a tedious and inefficient approach to MTL. A more automated way of weighting multiple tasks would thus be vastly appreciated.

In this experiment, we demonstrate how the DRPM can learn task difficulty by partitioning a network layer. Intuitively, a task that requires many neurons is more complex than a task that can be solved using a single neuron. Based on this observation, we propose the DRPM-MTL. The DRPM-MTL learns to partition the neurons of the last shared layer such that only a subset of the neurons are used for every task. In contrast to the other experiments (Sections 5.1 and 5.2), we use the DRPM without resampling and infer the partition $Y$ as a deterministic function. This can be done by applying the two-step procedure of Proposition 4.1 but skipping the resampling step of the MVHG and PL distributions. We compare the DRPM-MTL to the unitary loss scaling method (ULS, Kurin et al.,

2022), which has a fixed architecture and scales task losses equally. Both DRPM-MTL and ULS use a network with shared architecture up to some layer, after which the network branches into two task-specific layers that perform the classifications. Note the difference between the methods. While the task-specific branches of the ULS method access all neurons of the last shared layer, the task-specific branches of the DRPM-MTL access only the subset of neurons reserved for the respective task.

We perform experiments on MultiMNIST (Sabour et al., 2017), which overlaps two MNIST digits in one image, and we want to classify both numbers from a single sample. Hence, the two tasks, classification of the left and the right digit (see Appendix C.4 for an example), are approximately equal in difficulty by default. To increase the difficulty of one of the two tasks, we introduce the noisyMultiMNIST dataset. There, we control task difficulty by adding salt and pepper noise to one of the two digits, subsequently increasing the difficulty of that task with increasing noise ratios. Varying the noise, we evaluate how our DRPM-MTL adapts to imbalanced difficulties, where one usually has to tediously search for optimal loss weights to reach good performance. We base our pipeline on (Sener and Koltun, 2018). For more details and additional CelebA MTL experiments we refer to Appendix C.4.

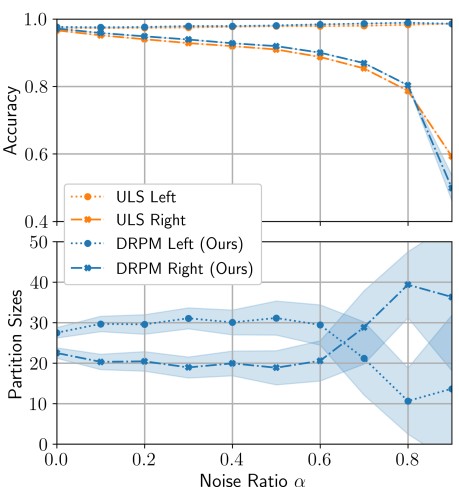

Figure 4: Results for noisyMultiMNIST experiment. In the upper plot, we compare the task accuracy of the two methods ULS and the DRPM-MTL. We see that the DRPM-MTL can reach higher accuracy for most of the different noise ratios $\alpha$ while it assigns the number of dimensions per task according to their difficulty.

We evaluate the DRPM-MTL concerning its classification accuracy on the two tasks and compare the inferred subset sizes per task for different noise ratios $\alpha \in \{0.0, \ldots, 0.9\}$ of the noisyMultiMNIST dataset (see Figure 4). The DRPM-MTL achieves the same or better accuracy on both tasks for most noise levels (upper part of Figure 4). It is interesting to see that, the more we increase $\alpha$, the more the DRPM-MTL tries to overcome the increased difficulty of the right task by assigning more dimensions to it (lower part of Figure 4, noise ratio $\alpha$ 0.6-0.8). Note that for the maximum noise ratio of $\alpha = 0.9$, it seems that the DRPM-MTL basically surrenders and starts neglecting the right task, instead focusing on getting good performance on the left task, which impacts the average accuracy.

## Limitations & Future Work

The proposed two-stage approach to RPMs requires distributions over subset sizes and permutation matrices. The memory usage of the permutation matrix used in the two-stage RPM increases quadratically in the number of elements $n$. Although we did not experience memory issues in our experiments, this may lead to problems when partitioning vast sets. Furthermore, learning subsets by first inferring an ordering of all elements can be a complex optimization problem. Approaches based on minimizing the earth mover's distance (Monge, 1781) to learn subset assignments could be an alternative to the ordering-based approach in our DRPM and pose an interesting direction for future work. Finally, note that we compute the probability mass function (PMF) $p(Y; \boldsymbol{\omega}, \boldsymbol{s})$ by approximating it with the bounds in Lemma 4.2. While the upper bound is tight when all scores have similar magnitude, the bound loosens if scores differ a lot, leading Equation (10) to overestimate the value of the PMF. In practice, we thus reweight the respective terms in the loss function, but in the future, we will investigate better estimates for the PMF.

Ultimately, we are interested in exploring how to apply the DRPM to multimodal learning under weak supervision, for instance, in medical applications. Section 5.2 demonstrated the potential of learning from coupled samples, but further research is needed to ensure fairness concerning underlying, hidden attributes when working with sensitive data.

## Conclusion

In this work, we proposed the differentiable random partition model, a novel approach to random partition models. Our two-stage method enables learning partitions end-to-end by separately controlling subset sizes and how elements are assigned to subsets. This new approach to partition learning enables the integration of random partition models into probabilistic and deterministic gradient-based optimization frameworks. We show the versatility of the proposed differentiable random partition model by applying it to three vastly different experiments. We demonstrate how learning partitions enables us to explore the modes of the data distribution, infer shared and independent generative factors from coupled samples, and learn task-specific sub-networks in applications where we want to solve multiple tasks on a single data point.

## Acknowledgements

AR is supported by the StimuLoop grant #1-007811-002 and the Vontobel Foundation. TS is supported by the grant #2021-911 of the Strategic Focal Area "Personalized Health and Related Technologies (PHRT)" of the ETH Domain (Swiss Federal Institutes of Technology).

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

# A    Preliminaries

## A.1    Hypergeometric Distribution

This part is largely based on Sutter et al. (2023).

Suppose we have an urn with marbles in different colors. Let $K \in \mathbb{N}$ be the number of different classes or groups (e.g. marble colors in the urn), $\boldsymbol{m} = [m_1, \ldots, m_K] \in \mathbb{N}^K$ describe the number of elements per class (e.g. marbles per color), $N = \sum_{k=1}^{K} m_K$ be the total number of elements (e.g. all marbles in the urn) and $n \in \{0, \ldots, N\}$ be the number of elements (e.g. marbles) to draw. Then, the multivariate hypergeometric distribution describes the probability of drawing $\boldsymbol{n} = [n_1, \ldots, n_K] \in \mathbb{N}^K$ marbles by sampling without replacement such that $\sum_{k=1}^{K} n_k = n$, where $n_k$ is the number of drawn marbles of class $k$.

In the literature, two different versions of the noncentral hypergeometric distribution exist, Fisher's (Fisher, 1935) and Wallenius' (Wallenius, 1963; Chesson, 1976) distribution. Sutter et al. (2023) restrict themselves to Fisher's noncentral hypergeometric distribution due to limitations of the latter (Fog, 2008). Hence, we will also talk solely about Fisher's noncentral hypergeometric distribution.

**Definition A.1** (Multivariate Fisher's Noncentral Hypergeometric Distribution (Fisher, 1935))**.** *A random vector $\boldsymbol{X}$ follows Fisher's noncentral multivariate distribution, if its joint probability mass function is given by*

$$P(\boldsymbol{N} = \boldsymbol{n}; \boldsymbol{\omega}) = p(\boldsymbol{n}; \boldsymbol{\omega}) = \frac{1}{P_0} \prod_{k=1}^{K} \binom{m_k}{n_k} \omega_k^{n_k} \tag{12}$$

$$\text{where} \quad P_0 = \sum_{(\eta_1, \ldots, \eta_K) \in \mathcal{S}} \prod_{k=1}^{K} \binom{m_k}{\eta_k} \omega_k^{\eta_k} \tag{13}$$

*The support $S$ of the PMF is given by $S = \{\boldsymbol{n} \in \mathbb{N}^K : \forall k \quad n_k \leq m_k, \sum_{k=1}^{K} n_k = n\}$ and $\binom{n}{k} = \frac{n!}{k!(n-k)!}$.*

The class importance $\boldsymbol{\omega}$ is a crucial modeling parameter in applying the noncentral hypergeometric distribution (see (Chesson, 1976)).

### A.1.1    Differentiable MVHG

Their reparameterizable sampling for the differentiable MVHG consists of three parts:

1. Reformulate the multivariate distribution as a sequence of interdependent and conditional univariate hypergeometric distributions.

2. Calculate the probability mass function of the respective univariate distributions.

3. Sample from the conditional distributions utilizing the Gumbel-Softmax trick.

Following the chain rule of probability, the MVHG distribution allows for sequential sampling over classes $k$. Every step includes a merging operation, which leads to biased samples compared to groundtruth non-differentiable sampling with equal class weights $\boldsymbol{\omega}$. Given that we intend to use the differentiable MVHG in settings where we want to learn the unknown class weights, we do not expect a negative effect from this sampling procedure. For details on how to merge the MVHG into a sequence of unimodal distributions, we refer to Sutter et al. (2023).

The probability mass function calculation is based on unnormalized log-weights, which are interpreted as unnormalized log-weights of a categorical distribution. The interpretation of the class-conditional unimodal hypergeometric distributions as categorical distributions allows applying the Gumbel-Softmax trick (Jang et al., 2016; Maddison et al., 2017). Following the use of the Gumbel-Softmax trick, the class-conditional version of the hypergeometric distribution is differentiable and reparameterizable. Hence, the MVHG has been made differentiable and reparameterizable as well. Again, for details we refer to the original paper (Sutter et al., 2023).

## A.2 Distribution over Random Orderings

Yellott (1977) show that the distribution over permutation matrices $p(\pi; \boldsymbol{s})$ follows a Plackett-Luce (PL) distribution (Plackett, 1975; Luce, 1959), if and only of the perturbed scores $\tilde{\boldsymbol{s}}$ are sampled independently from Gumbel distributions with identical scales. For each item $i$, sample $g_i \sim \text{Gumbel}(0, \beta)$ independently with zero mean and and fixed scale $\beta$. Let $\tilde{\boldsymbol{s}}$ be the vector of Gumbel perturbed log-weights such that $\tilde{s}_i = \beta \log s_i + g_i$. Hence,

$$q(\tilde{s}_1 \geq \cdots \geq \tilde{s}_n) = \frac{s_1}{Z} \cdot \frac{s_2}{Z - s_1} \cdot \ldots \cdot \frac{s_n}{Z - \sum_{i=1}^{n-1} s_i} \tag{14}$$

We refer to Yellott (1977) or Grover et al. (2019) for the proof. However, Grover et al. (2019) provide only an adapted proof sketch from Yellott (1977). The probability of sampling element $i$ first is given by its score $s_i$ divided by the sum of all weights in the set

$$q(\tilde{s}_i) = \frac{s_i}{Z} \tag{15}$$

For $z_i = \log s_i$, the right hand side of Equation (15) is equal to the softmax distribution $\text{softmax}(z_i) = \exp(z_i)/\sum_j \exp(z_j)$ as already described in (Xie and Ermon, 2019). Hence, Equation (15) directly leads to the Gumbel-Softmax trick (Jang et al., 2016; Maddison et al., 2017).

### A.2.1 Differentiable Sorting

In the main text of the paper we rely on a differentiable function $f_\pi(\tilde{\boldsymbol{s}})$, which sorts the resampled version of the scores $\boldsymbol{s}$

$$\pi = f_\pi(\tilde{\boldsymbol{s}}) = \text{sort}(\tilde{\boldsymbol{s}}) \tag{16}$$

Here, we summarise the findings from Grover et al. (2019) on how to construct such a differentiable sorting operator. As already mentioned in Section 2, there are multiple works on the topic (Prillo and Eisenschlos, 2020; Petersen et al., 2021; Mena et al., 2018), but we restrict ourselves to the work of Grover et al. (2019) as we see the differentiable generation of permutation matrices as a tool in our pipeline.

**Corollary A.2** (Permutation Matrix (Grover et al., 2019)). *Let $\boldsymbol{s} = [s_1, \ldots, s_n]^T$ be a real-valued vector of length $n$. Let $A_{\boldsymbol{s}}$ denote the matrix of absolute pairwise differences of the elements of $\boldsymbol{s}$ such that $A_{\boldsymbol{s}}[i, j] = |s_i - s_j|$. The permutation matrix $\pi$ corresponding to sort($\boldsymbol{s}$) is given by:*

$$\pi = \begin{cases} 1 & \text{if } j = \arg\max[(n + 1 - 2i)\boldsymbol{s} - A_{\boldsymbol{s}}\mathbb{1}] \\ 0 & \text{otherwise} \end{cases} \tag{17}$$

*where $\mathbb{1}$ denotes the column vector of all ones.*

As we know, the $\arg\max$ operator is non-differentiable which prohibits the direct use of Corollary A.2 for gradient computation. Hence, Grover et al. (2019) propose to replace the $\arg\max$ operator with softmax to obtain a continuous relaxation $\pi(\tau)$ similar to the GS trick (Jang et al., 2016; Maddison et al., 2017). In particular, the $i$th row of $\pi(\tau)$ is given by:

$$\pi(\tau)[i, :] = \text{softmax}[(n + 1 - 2i)\boldsymbol{s} - A_{\boldsymbol{s}}\mathbb{1}/\tau] \tag{18}$$

where $\tau > 0$ is a temperature parameter. We adapted this section from Grover et al. (2019) and we also refer to their original work for more details on how to generate differentiable permutation matrices.

In this, work we remove the temperature parameter $\tau$ to reduce clutter in the notation. Hence, we only write $\pi$ instead of $\pi(\tau)$, although it is still needed for the generation of the matrix $\pi$. For details on how we select the temperature parameter $\tau$ in our experiments, we refer to Appendix C.

## B Detailed Derivation of the Differentiable Two-Stage Random Partition Model

### B.1 Two-Stage Partition Model

We want to partition $n$ elements $[n] = \{1, \ldots, n\}$ into $K$ subsets $\{\mathcal{S}_1, \ldots, \mathcal{S}_K\}$ where $K$ is *a priori* unknown.

**Definition B.1** (Partition). *A partition $\rho$ of a set of elements $[n] = \{1, \ldots, n\}$ is a collection of subsets $(\mathcal{S}_1, \ldots, \mathcal{S}_K)$ such that*

$$\mathcal{S}_1 \cup \cdots \cup \mathcal{S}_K = [n] \quad and \quad \forall i \neq j : \ \mathcal{S}_i \cap \mathcal{S}_j = \emptyset \tag{19}$$

Put differently, every element $i$ has to be assigned to precisely one subset $\mathcal{S}_k$. We denote the size of the $k$-th subset $\mathcal{S}_k$ as $n_k = |\mathcal{S}_k|$. Alternatively, we describe a partition $\rho$ as an assignment matrix $Y = [\boldsymbol{y}_1, \ldots, \boldsymbol{y}_K]^T \in \{0, 1\}^{K \times n}$. Every row $\boldsymbol{y}_k \in \{0, 1\}^{1 \times n}$ is a multi-hot vector, where $\boldsymbol{y}_{ki} = 1$ assigns element $i$ to subset $\mathcal{S}_k$.

In this work, we propose a new two-stage procedure to learn partitions. The proposed formulation separately infers the number of elements per subset $n_k$ and the assignment of elements to subsets $\mathcal{S}_k$ by inducing an order on the $n$ elements and filling $\mathcal{S}_1, ..., \mathcal{S}_K$ sequentially in this order. See Figure 1 for an example.

**Definition B.2** (Two-stage partition model). *Let $\boldsymbol{n} = [n_1, \ldots, n_K] \in \mathbb{N}_0^K$ be the subset sizes in $\rho$, with $\mathbb{N}_0$ the set of natural numbers including 0 and $\sum_{k=1}^K n_k = n$, where $n$ is the total number of elements. Let $\pi \in \{0, 1\}^{n \times n}$ be a permutation matrix that defines an order over the $n$ elements. We define the two-stage partition model of $n$ elements into $K$ subsets as an assignment matrix $Y = [\boldsymbol{y}_1, \ldots, \boldsymbol{y}_K]^T \in \{0, 1\}^{K \times n}$ with*

$$\boldsymbol{y}_k = \sum_{i=\nu_k+1}^{\nu_k + n_k} \boldsymbol{\pi}_i, \quad where \quad \nu_k = \sum_{\iota=1}^{k-1} n_\iota \tag{20}$$

*such that $Y = [\{\boldsymbol{y}_k \mid n_k > 0\}_{k=1}^K]^T$.*

Note that in contrast to previous work on partition models (Mansour and Schork, 2016), we allow $\mathcal{S}_k$ to be the empty set $\emptyset$. Hence, $K$ defines the maximum number of possible subsets, not the effective number of non-empty subsets.

To model the order of the elements, we use a permutation matrix $\pi = [\boldsymbol{\pi}_1, \ldots, \boldsymbol{\pi}_n]^T \in \{0, 1\}^{n \times n}$ which is a square matrix where every row and column sums to 1. This doubly-stochastic property of all permutation matrices $\pi$ (Marcus, 1960) thus ensures that the columns of $Y$ remain one-hot vectors. At the same time, its rows correspond to $n_k$-hot vectors $\boldsymbol{y}_k$ in Definition B.2 and therefore serve as subset assignment vectors.

**Corollary B.3.** *A two-stage partition model $Y$, which follows Definition B.2, is a valid partition satisfying Definition B.1.*

*Proof.* By definition, every row $\boldsymbol{\pi}_i$ and column $\boldsymbol{\pi}_j$ of $\pi$ is a one-hot vector, hence every $\sum_{i=\nu_k+1}^{\nu_k+n_k} \boldsymbol{\pi}_i$ results in different, non-overlapping $n_k$-hot encodings, ensuring $\mathcal{S}_i \cap \mathcal{S}_j = \emptyset \ \forall \ i, j$ and $i \neq j$. Further, since $n_k$-hot encodings have exactly $n_k$ entries with 1, we have $\sum_{i=\nu_k+1}^{\nu_k+n_k} \sum_{j=1}^n \boldsymbol{\pi}_{ij} = n_k$. Hence, since $\sum_{k=1}^K n_k = n$, every element $i$ is assigned to a $\boldsymbol{y}_k$, ensuring $\mathcal{S}_1 \cup \cdots \cup \mathcal{S}_K = [n]$. $\square$

### B.2 Two-Stage Random Partition Models

An RPM $p(Y)$ defines a probability distribution over partitions $Y$. In this section, we derive how to extend the two-stage procedure from Definition B.2 to the probabilistic setting to create a two-stage RPM. To derive the two-stage RPM's probability distribution $p(Y)$, we need to model distributions over $\boldsymbol{n}$ and $\pi$. We choose the MVHG distribution $p(\boldsymbol{n}; \boldsymbol{\omega})$ and the PL distribution $p(\pi; \boldsymbol{s})$ (see Section 3).

We calculate the probability $p(Y; \boldsymbol{\omega}, \boldsymbol{s})$ sequentially over the probabilities of subsets $p_{\boldsymbol{y}_k} := p(\boldsymbol{y}_k \mid \boldsymbol{y}_{<k}; \boldsymbol{\omega}, \boldsymbol{s})$. $p_{\boldsymbol{y}_k}$ itself depends on the probability over subset permutations $p_{\bar{\pi}_k} := p(\bar{\pi} \mid n_k, \boldsymbol{y}_{<k}; \boldsymbol{s})$, where a subset permutation matrix $\bar{\pi}$ represents an ordering over $n_k$ out of $n$ elements.

**Definition B.4** (Subset permutation matrix $\bar{\pi}$). *A subset permutation matrix $\bar{\pi} \in \{0, 1\}^{n_k \times n}$, where $n_k \leq n$, must fulfill*

$$\forall i \leq n_k : \ \sum_{j=1}^n \bar{\pi}_{ij} = 1 \quad and \quad \forall j \leq n : \ \sum_{i=1}^{n_k} \bar{\pi}_{ij} \leq 1.$$

We describe the probability distribution over subset permutation matrices $p_{\bar{\pi}_k}$ using Definition B.4 and Equation (3).

**Lemma B.5** (Probability over subset permutations $p_{\bar{\pi}_k}$). *The probability $p_{\bar{\pi}_k}$ of any subset permutation matrix $\bar{\pi} = [\bar{\pi}_1, \ldots, \bar{\pi}_{n_k}]^T \in \{0,1\}^{n_k \times n}$ is given by*

$$p_{\bar{\pi}_k} := p(\bar{\pi} \mid n_k, \boldsymbol{y}_{<k}; \boldsymbol{s}) = \prod_{i=1}^{n_k} \frac{(\bar{\pi}\boldsymbol{s})_i}{Z_k - \sum_{j=1}^{i-1}(\bar{\pi}\boldsymbol{s})_j} \tag{21}$$

*where $\boldsymbol{y}_{<k} = \{\boldsymbol{y}_1, ..., \boldsymbol{y}_{k-1}\}$, $Z_k = Z - \sum_{j \in \mathcal{S}_{<k}} s_j$ and $\mathcal{S}_{<k} = \bigcup_{j=1}^{k-1} \mathcal{S}_j$.*

*Proof.* We provide the proof for $p_{\bar{\pi}_1}$, but it is equivalent for all other subsets. Without loss of generality, we assume that there are $n_1$ elements in $\mathcal{S}_1$. Following Equation (3), the probability of a permutation matrix $p(\pi; \boldsymbol{s})$ is given by

$$p(\pi; \boldsymbol{s}) = \frac{(\pi\boldsymbol{s})_1}{Z} \frac{(\pi\boldsymbol{s})_2}{Z - (\pi\boldsymbol{s})_1} \cdots \frac{(\pi\boldsymbol{s})_n}{Z - \sum_{j=1}^{n-1}(\pi\boldsymbol{s})_j} \tag{22}$$

At the moment, we are only interested in the ordering of the first $n_1$ elements. The probability of the first $n_1$ is given by marginalizing over the remaining $n - n_1$ elements:

$$p(\bar{\pi} \mid n_1; \boldsymbol{\omega}) = \sum_{\pi \in \Pi_1} p(\pi \mid \boldsymbol{s}) \tag{23}$$

where $\Pi_1$ is the set of permutation matrices such that the top $n_1$ rows select the elements in a specific ordering $\bar{\pi} \in \{0,1\}^{n_1 \times n}$, i.e. $\Pi_1 = \{\pi : [\boldsymbol{\pi}_1, \ldots, \boldsymbol{\pi}_{n_1}]^T = \bar{\pi}\}$. It follows

$$p(\bar{\pi} \mid n_1; \boldsymbol{\omega}) = \sum_{\pi \in \Pi_1} p(\pi \mid \boldsymbol{s}) \tag{24}$$

$$= \sum_{\pi \in \Pi_1} \prod_{i=1}^{n} \frac{(\pi\boldsymbol{s})_i}{Z - \sum_{j=1}^{i-1}(\pi\boldsymbol{s})_j} \tag{25}$$

$$= \prod_{i=1}^{n_1} \frac{(\bar{\pi}\boldsymbol{s})_i}{Z - \sum_{j=1}^{i-1}(\bar{\pi}\boldsymbol{s})_j} \sum_{\pi \in \Pi_1} \prod_{i=1}^{n-n_1} \frac{(\pi\boldsymbol{s})_{n_1+i}}{Z - \sum_{j=1}^{n_1}(\bar{\pi}\boldsymbol{s})_j - \sum_{j=1}^{i-1}(\bar{\pi}\boldsymbol{s})_j} \tag{26}$$

$$= \prod_{i=1}^{n_1} \frac{(\bar{\pi}\boldsymbol{s})_i}{Z - \sum_{j=1}^{i-1}(\bar{\pi}\boldsymbol{s})_j} \sum_{\pi \in \Pi_1} \prod_{i=1}^{n-n_1} \frac{(\pi\boldsymbol{s})_{n_1+i}}{Z_1 - \sum_{j=1}^{i-1}(\bar{\pi}\boldsymbol{s})_j} \tag{27}$$

where $Z_1 = Z - \sum_{j=1}^{n_1}(\bar{\pi}\boldsymbol{s})_j$. It follows

$$p(\bar{\pi} \mid n_1; \boldsymbol{\omega}) = \prod_{i=1}^{n_1} \frac{(\bar{\pi}\boldsymbol{s})_i}{Z - \sum_{j=1}^{i-1}(\bar{\pi}\boldsymbol{s})_j} \tag{28}$$

$\square$

Lemma B.5 describes the probability of drawing the elements $i \in \mathcal{S}_k$ in the order described by the subset permutation matrix $\bar{\pi}$ given that the elements in $\mathcal{S}_{<k}$ are already determined. Note that in a slight abuse of notation, we use $p(\bar{\pi} \mid n_k, \boldsymbol{y}_{<k}; \boldsymbol{\omega}, \boldsymbol{s})$ as the probability of a subset permutation $\bar{\pi}$ given that there are $n_k$ elements in $\mathcal{S}_k$ and thus $\bar{\pi} \in \{0,1\}^{n_k \times n}$. Additionally, we condition on the subsets $\boldsymbol{y}_{<k}$ and $n_k$, the size of subset $\mathcal{S}_k$. In contrast to the distribution over permutations matrices $p(\pi; \boldsymbol{s})$ in Equation (3), we take the product over $n_k$ terms and have a different normalization constant $Z_k$. Although we induce an ordering over all elements $i$ in Definition B.2, the probability $p_{\boldsymbol{y}_k}$ is invariant to intra-subset orderings of elements $i \in \mathcal{S}_k$.

**Lemma B.6** (Probability distribution $p_{\boldsymbol{y}_k}$). *The probability distribution over subset assignments $p_{\boldsymbol{y}_k}$ is given by*

$$p_{\boldsymbol{y}_k} := p(\boldsymbol{y}_k \mid \boldsymbol{y}_{<k}; \boldsymbol{\omega}, \boldsymbol{s}) = p(n_k \mid n_{<k}; \boldsymbol{\omega}) \sum_{\bar{\pi} \in \Pi_{\boldsymbol{y}_k}} p(\bar{\pi} \mid n_k, \boldsymbol{y}_{<k}; \boldsymbol{s})$$

*where $\Pi_{\boldsymbol{y}_k} = \{\bar{\pi} \in \{0,1\}^{n_k \times n} : \boldsymbol{y}_k = \sum_{i=1}^{n_k} \bar{\pi}_i\}$ and $p(\bar{\pi} \mid n_k, \boldsymbol{y}_{<k}; \boldsymbol{s})$ as in Lemma B.5.*

*Proof.* We can proof the statement of Lemma B.6 as follows:

$$p_{\boldsymbol{y}_k} = p(\boldsymbol{y}_k \mid \boldsymbol{y}_{<k}; \boldsymbol{\omega}, \boldsymbol{s})$$

$$= \sum_{n_k'} p(\boldsymbol{y}_k, n_k' \mid \boldsymbol{y}_{<k}; \boldsymbol{\omega}, \boldsymbol{s}) \tag{29}$$

$$= \sum_{n_k'} p(n_k' \mid \boldsymbol{y}_{<k}; \boldsymbol{\omega}, \boldsymbol{s}) p(\boldsymbol{y}_k \mid n_k', \boldsymbol{y}_{<k}; \boldsymbol{\omega}, \boldsymbol{s}) \tag{30}$$

$$= \sum_{n_k'} p(n_k' \mid n_{<k}; \boldsymbol{\omega}, \boldsymbol{s}) p(\boldsymbol{y}_k \mid n_k', \boldsymbol{y}_{<k}; \boldsymbol{s}) \tag{31}$$

$$= p(n_k \mid n_{<k}; \boldsymbol{\omega}, \boldsymbol{s}) p(\boldsymbol{y}_k \mid n_k, \boldsymbol{y}_{<k}; \boldsymbol{s}) \tag{32}$$

$$= p(n_k \mid n_{<k}; \boldsymbol{\omega}) \sum_{\bar{\pi} \in \Pi_{\boldsymbol{y}_k}} p(\bar{\pi} \mid n_k, \boldsymbol{y}_{<k}; \boldsymbol{s}) \tag{33}$$

Equation (29) holds by marginalization, where $n_k'$ denotes the random variable that stands for the size of subset $\mathcal{S}_k$. By Bayes' rule, we can then derive Equation (30). The next derivations stem from the fact that we can compute $n_{<k}$ if $\boldsymbol{y}_{<k}$ is given, as the assignments $\boldsymbol{y}_{<k}$ hold information on the size of subsets $\mathcal{S}_{<k}$. More explicitly, $n_i = \sum_{j=1}^n y_{ij}$. Further, $\boldsymbol{y}_k$ is independent of $\boldsymbol{\omega}$ if the size $n_k'$ of subset $\mathcal{S}_k$ is given, leading to Equation (31). We further observe that $p(\boldsymbol{y}_k \mid n_k', \boldsymbol{y}_{<k}; \boldsymbol{s})$ is only non-zero, if $n_k' = \sum_{i=1}^n y_{ki} = n_k$. Dropping all zero terms from the sum in Equation (31) thus results in Equation (32). Finally, by Definition B.2, we know that $\boldsymbol{y}_k = \sum_{i=\nu_k+1}^{\nu_k+n_k} \boldsymbol{\pi}_i$, where $\nu_k = \sum_{\iota=1}^{k-1} n_\iota$ and $\pi \in \{0,1\}^{n \times n}$ a permutation matrix. Hence, in order to get $\boldsymbol{y}_k$ given $\boldsymbol{y}_{<k}$, we need to marginalize over all permutations of the elements of $\boldsymbol{y}_k$ given that the elements in $\boldsymbol{y}_{<k}$ are already ordered, which corresponds exactly to marginalizing over all subset permutation matrices $\bar{\pi}$ such that $\boldsymbol{y}_k = \sum_{i=1}^{n_k} \bar{\boldsymbol{\pi}}_i$, resulting in Equation (33). $\square$

In Lemma B.6, we describe the set of all subset permutations $\bar{\pi}$ of elements $i \in \mathcal{S}_k$ by $\Pi_{\boldsymbol{y}_k}$. Put differently, we make $p(\boldsymbol{y}_k \mid \boldsymbol{y}_{<k}; \boldsymbol{\omega}, \boldsymbol{s})$ invariant to the ordering of elements $i \in \mathcal{S}_k$ by marginalizing over the probabilities of subset permutations $p_{\bar{\pi}_k}$ (Xie and Ermon, 2019).

Using Lemmas B.5 and B.6, we propose the two-stage random partition $p(Y; \boldsymbol{\omega}, \boldsymbol{s})$. Since $Y = [\boldsymbol{y}_1, \ldots, \boldsymbol{y}_K]^T$, we calculate $p(Y; \boldsymbol{\omega}, \boldsymbol{s})$, the PMF of the two-stage RPM, sequentially using Lemmas B.5 and B.6, where we leverage the PL distribution for permutation matrices $p(\pi; \boldsymbol{s})$ to describe the probability distribution over subsets $p(\boldsymbol{y}_k \mid \boldsymbol{y}_{<k}; \boldsymbol{\omega}, \boldsymbol{s})$.

**Proposition 4.1** (Two-Stage Random Partition Model). Given a probability distribution over subset sizes $p(\boldsymbol{n}; \boldsymbol{\omega})$ with $\boldsymbol{n} \in \mathbb{N}_0^K$ and distribution parameters $\boldsymbol{\omega} \in \mathbb{R}_+^K$ and a PL probability distribution over random orderings $p(\pi; \boldsymbol{s})$ with $\pi \in \{0,1\}^{n \times n}$ and distribution parameters $\boldsymbol{s} \in \mathbb{R}_+^n$, the probability mass function $p(Y; \boldsymbol{\omega}, \boldsymbol{s})$ of the two-stage RPM is given by

$$p(Y; \boldsymbol{\omega}, \boldsymbol{s}) = p(\boldsymbol{y}_1, \ldots, \boldsymbol{y}_K; \boldsymbol{\omega}, \boldsymbol{s}) = p(\boldsymbol{n}; \boldsymbol{\omega}) \sum_{\pi \in \Pi_Y} p(\pi; \boldsymbol{s}) \tag{34}$$

where $\Pi_Y = \{\pi : \boldsymbol{y}_k = \sum_{i=\nu_k+1}^{\nu_k+n_k} \boldsymbol{\pi}_i, k = 1, \ldots, K\}$, and $\boldsymbol{y}_k$ and $\nu_k$ as in Definition B.2.

*Proof.* Using Lemmas B.5 and B.6, we write

$$
\begin{aligned}
p(Y) =& p(\boldsymbol{y}_1, \dots, \boldsymbol{y}_K; \boldsymbol{\omega}, \boldsymbol{s}) = p(\boldsymbol{y}_1; \boldsymbol{\omega}, \boldsymbol{s}) \cdots p(\boldsymbol{y}_K \mid \{\boldsymbol{y}_j\}_{j<K}; \boldsymbol{\omega}, \boldsymbol{s}) \\
=& \left( p(n_1; \boldsymbol{\omega}) \sum_{\bar{\pi}_1 \in \Pi_{\boldsymbol{y}_1}} p(\bar{\pi}_1 \mid n_1; \boldsymbol{s}) \right) \\
& \cdots \left( p(n_K \mid \{n_j\}_{j<K}; \boldsymbol{\omega}) \sum_{\bar{\pi}_K \in \Pi_{\boldsymbol{y}_K}} p(\bar{\pi}_K \mid \{n_j\}_{j \le K}; \boldsymbol{s}) \right) \quad (35) \\
=& p(n_1; \boldsymbol{\omega}) \cdots p(n_K \mid \{n_K\}_{j<K}; \boldsymbol{\omega}) \\
& \cdot \left( \sum_{\bar{\pi}_1 \in \Pi_{\boldsymbol{y}_1}} p(\bar{\pi}_1 \mid n_1; \boldsymbol{s}) \cdots \sum_{\pi_K \in \Pi_{\boldsymbol{y}_K}} p(\bar{\pi}_K \mid \{n_j\}_{j \le K}; \boldsymbol{s}) \right) \quad (36) \\
=& p(\boldsymbol{n}; \boldsymbol{\omega}) \left( \sum_{\bar{\pi}_1 \in \Pi_{\boldsymbol{y}_1}} \cdots \sum_{\pi_K \in \Pi_{\boldsymbol{y}_K}} p(\bar{\pi}_1 \mid n_1; \boldsymbol{s}) \cdots p(\bar{\pi}_K \mid \{n_j\}_{j \le K}; \boldsymbol{s}) \right) \quad (37) \\
=& p(\boldsymbol{n}; \boldsymbol{\omega}) \sum_{\pi \in \Pi_Y} p(\pi \mid \boldsymbol{n}; \boldsymbol{s}) \quad (38) \\
=& p(\boldsymbol{n}; \boldsymbol{\omega}) \sum_{\pi \in \Pi_Y} p(\pi; \boldsymbol{s}) \quad (39)
\end{aligned}
$$

$\square$

## B.3 Approximating the Probability Mass Function

**Lemma 4.2.** $p(Y; \boldsymbol{\omega}, \boldsymbol{s})$ can be upper and lower bounded as follows

$$
\forall \pi \in \Pi_Y : \ p(\boldsymbol{n}; \boldsymbol{\omega}) p(\pi; \boldsymbol{s}) \ \le \ p(Y; \boldsymbol{\omega}, \boldsymbol{s}) \ \le \ |\Pi_Y| p(\boldsymbol{n}; \boldsymbol{\omega}) \max_{\tilde{\pi}} p(\tilde{\pi}; \boldsymbol{s}) \quad (40)
$$

*Proof.* Since $p(\pi; \boldsymbol{s})$ is a probability we know that $\forall \pi \in \{0, 1\}^{n \times n} \ \ p(\pi; \boldsymbol{s}) \ge 0$. Thus, it follows directly that:

$$
\forall \pi \in \Pi_Y : \ \ p(Y; \boldsymbol{\omega}, \boldsymbol{s}) = p(\boldsymbol{n}; \boldsymbol{\omega}) \sum_{\pi' \in \Pi_Y} p(\pi'; \boldsymbol{s}) \ge p(\boldsymbol{n}; \boldsymbol{\omega}) p(\pi; \boldsymbol{s}),
$$

proving the lower bound of Lemma 4.2.

On the other hand, can prove the upper bound in Lemma 4.2 by:

$$
\begin{aligned}
p(Y; \boldsymbol{\omega}, \boldsymbol{s}) &= p(\boldsymbol{n}; \boldsymbol{\omega}) \sum_{\pi' \in \Pi_Y} p(\pi'; \boldsymbol{s}) \\
&\le p(\boldsymbol{n}; \boldsymbol{\omega}) \sum_{\pi' \in \Pi_Y} \max_{\pi \in \Pi_Y} p(\pi; \boldsymbol{s}) \\
&= p(\boldsymbol{n}; \boldsymbol{\omega}) \max_{\pi \in \Pi_Y} p(\pi; \boldsymbol{s}) \sum_{\pi' \in \Pi_Y} 1 \\
&= |\Pi_Y| \cdot p(\boldsymbol{n}; \boldsymbol{\omega}) \max_{\pi \in \Pi_Y} p(\pi; \boldsymbol{s}) \\
&\le |\Pi_Y| \cdot p(\boldsymbol{n}; \boldsymbol{\omega}) \max_{\pi} p(\pi; \boldsymbol{s})
\end{aligned}
$$

We can compute the maximum probability $\max_\pi p(\pi; \boldsymbol{s})$ with the probability of the permutation matrix $f_\pi(\boldsymbol{s})$, which sorts the unperturbed scores in decreasing order. $\square$

## B.4 The Differentiable Random Partition Model

We propose the DRPM $p(Y; \boldsymbol{\omega}, \boldsymbol{s})$, a differentiable and reparameterizable two-stage RPM.

**Lemma 4.3** (DRPM). A two-stage RPM is differentiable and reparameterizable if the distribution over subset sizes $p(\boldsymbol{n}; \boldsymbol{\omega})$ and the distribution over orderings $p(\pi; \boldsymbol{s})$ are differentiable and reparameterizable.

*Proof.* To prove that our two-stage RPM is differentiable we need to prove that we can compute gradients for the bounds in Lemma 4.2 and to provide a reparameterization scheme for the two-stage approach in Definition B.2.

**Gradients for the bounds:** Since we assume that $p(\boldsymbol{n}; \boldsymbol{\omega})$ and $p(\pi; \boldsymbol{s})$ are differentiable and reparameterizable, we only need to show that we can compute $|\Pi_Y|$ and $\max_{\tilde{\pi}} p(\tilde{\pi}; \boldsymbol{s})$ in a differentiable manner to prove that the bounds in Lemma 4.2 are differentiable. By definition (see Section 4.1),

$$|\Pi_Y| = \prod_{k=1}^{K} |\Pi_{\boldsymbol{y}_k}| = \prod_{k=1}^{K} n_k!.$$

Hence, $|\Pi_Y|$ can be computed given a reparametrized version $n_k$, which is provided by the reparametrization trick for the MVHG $p(\boldsymbol{n}; \boldsymbol{\omega})$. Further, from Equation (14), we immediately see that the most probable permutation is given by the order induced by sorting the original, unperturbed scores $\boldsymbol{s}$ from highest to lowest. This implies that $\max_{\tilde{\pi}} p(\tilde{\pi}; \boldsymbol{s}) = p(\pi_{\boldsymbol{s}}; \boldsymbol{s})$, which we can compute due to $p(\pi_{\boldsymbol{s}}; \boldsymbol{s})$ being differentiable according to our assumptions.

**Reparametrization of the two-stage approach:** Given reparametrized versions of $\boldsymbol{n}$ and $\pi$, we compute a partition as follows:

$$\boldsymbol{y}_k = \sum_{i=\nu_k+1}^{\nu_k+n_k} \boldsymbol{\pi}_i, \quad \text{where} \quad \nu_k = \sum_{\iota=1}^{k-1} n_\iota \tag{41}$$

The challenge here is that we need to be able to backpropagate through $n_k$, which appears as an index in the sum. Let $\boldsymbol{\alpha}_k = \{0, 1\}^n$, such that

$$(\boldsymbol{\alpha}_k)_i = \begin{cases} 1 & \text{if } \nu_k < i \le \nu_{k+1} \\ 0 & \text{otherwise} \end{cases}$$

Given such $\boldsymbol{\alpha}_k$, we can rewrite Equation (41) with

$$\boldsymbol{y}_k = \sum_{i=1}^{n} (\boldsymbol{\alpha}_k)_i \boldsymbol{\pi}_i. \tag{42}$$

While this solves the problem of propagating through sum indices, it is not clear how to compute $\boldsymbol{\alpha}_k$ in a differentiable manner. Similar to other works on continuous relaxations (Jang et al., 2016; Maddison et al., 2017), we can compute a relaxation of $\boldsymbol{\alpha}_k$ by introducing a temperature $\tau$. Let us introduce auxiliary function $f : \mathbb{N} \to [0, 1]^n$, that maps an integer $x$ to a vector with entries

$$f_i(x; \tau) = \sigma\left(\frac{x - i + \epsilon}{\tau}\right),$$

such that $f_i(x; \tau) \approx 0$ if $\frac{x-i}{\tau} < 0$ and $f_i(x; \tau) \approx 1$ if $\frac{x-i}{\tau} \ge 0$. Note that $\sigma(\cdot)$ is the standard sigmoid function and $\epsilon << 1$ is a small positive constant to break the tie at $\sigma(0)$. We then compute an approximation of $\boldsymbol{\alpha}_k$ with

$$\tilde{\boldsymbol{\alpha}}_k(\tau) = f(\nu_k; \tau) - f(\nu_{k-1}; \tau),$$

$\tilde{\boldsymbol{\alpha}}_k(\tau) \in [0, 1]^n$. Then, for $\tau \to 0$ we have $\tilde{\boldsymbol{\alpha}}_k(\tau) \to \boldsymbol{\alpha}_k$. In practice, we cannot set $\tau = 0$ since this would amount to a division by 0. Instead, we can apply the straight-through estimator (Bengio et al., 2013) to the auxiliary function $f(x; \tau)$ in order to get $\tilde{\boldsymbol{\alpha}}_k \in \{0, 1\}^n$ and use it to compute Equation (42). $\square$

Note that in our experiments, we use the MVHG relaxation of Sutter et al. (2023) and can thus leverage that they return one-hot encodings for $n_k$. This allows a different path for computing $\boldsymbol{\alpha}_k$ which circumvents introducing yet another temperature parameter altogether. We refer to our code in the supplement for more details.

Table 3: Total GPU hours per experiment. We report the cumulative training and testing hours to generate the results shown in the main part of this manuscript. We relied on our internal cluster infrastructure equipped with RTX2080Ti GPUs. Hence, we report the number of compute hours for this GPU-type.

| Experiment | Computation Time (h) |
|---|---|
| Clustering (Section 5.1) | 100 |
| Partitioning of Generative Factors (Section 5.2) | 480 |
| MTL (Section 5.3) | 100 |

## C  Experiments

In the following, we describe each of our experiments in more detail and provide additional ablations. All our experiments were run on RTX2080Ti GPUs. Each run took 6h-8h (Variational Clustering), 4h-6h (Generative Factor Partitioning), or $\sim$ 1h (Multitask Learning) respectively. We report the training and test time per model. Please note that we can only report the numbers to generate the final results but not the development time.

**Code Release**  The official code can be found under `https://github.com/thomassutter/drpm`. Please note that the results reported in the main text slightly differ from the ones being generated from the official code. For the main paper, we based our own code for the experiments in Section 5.2 on the `disentanglement_lib` (Locatello et al., 2020). However, the library is based on Tensorflow v1 (Abadi et al., 2016), which makes it more and more difficult to maintain and install. Therefore, we decided to re-implement everything in PyTorch (Paszke et al., 2019).

While the metrics of our method and the baselines slightly change, the relative performance between them remains the same.

The code and results for the remaining two experiments in Sections 5.1 and 5.3 are the same as in the main text.

### C.1  Approximation quality of Lemma 4.2

To provide intuitive understanding of the bounds introduced in Lemma 4.2, we present an experiment in this subsection to demonstrate the behavior of the upper and lower bounds. It is important to note that RPMs are discrete distributions, as the number of possible samples is finite for a given number of elements $n$ and subsets $K$. Therefore, we can estimate the probability of a fixed partition $\tilde{Y}$ under given $\boldsymbol{\omega}$ and $\boldsymbol{s}$ by sampling $M$ partitions from $p(Y; \boldsymbol{\omega}, \boldsymbol{s})$, counting the occurrences of $\tilde{Y}$ in the samples, and dividing the count by $M$. As $M$ approaches infinity, we can obtain the true probability mass function (PMF) $p(Y; \boldsymbol{\omega}, \boldsymbol{s})$ for every partition $Y$.

In our experiment, we set $n = 5$ and aim to evaluate the quality of our bounds for all possible subset combinations of 5 elements, we thus set $K = n = 5$. To obtain a reliable estimate of the true PMF, we set $M = 10^8$. From Lemma 4.2, we know that:

$$\forall \pi \in \Pi_Y : \ p(\boldsymbol{n}; \boldsymbol{\omega})p(\pi; \boldsymbol{s}) \ \leq \ p(Y; \boldsymbol{\omega}, \boldsymbol{s}) \ \leq \ |\Pi_Y|p(\boldsymbol{n}; \boldsymbol{\omega}) \max_{\tilde{\pi}} p(\tilde{\pi}; \boldsymbol{s})$$

Let us define

$$p_U(Y; \boldsymbol{\omega}, \boldsymbol{s}) := |\Pi_Y|p(\boldsymbol{n}; \boldsymbol{\omega}) \max_{\tilde{\pi}} p(\tilde{\pi}; \boldsymbol{s})$$

$$p_L(Y; \boldsymbol{\omega}, \boldsymbol{s}) := \max_{\pi \in \Pi_Y} p(\boldsymbol{n}; \boldsymbol{\omega})p(\pi; \boldsymbol{s})$$

In Figure 5, we present the estimated PMF along with the corresponding upper bounds ($p_U$) and lower bounds ($p_L$) for four different combinations of RPM parameters $\boldsymbol{\omega}$ and $\boldsymbol{s}$. We observe that when all scores $\boldsymbol{s}$ are equal, as in the priors of the experiments in Sections 5.1 and 5.2, $p_U(Y; \boldsymbol{\omega}, \boldsymbol{s})$ approximates $p(Y; \boldsymbol{\omega}, \boldsymbol{s})$ well and serves as a reliable estimate of the PMF. However, when the scores vary, the upper bound becomes looser, particularly for lower probability partitions, as it is dominated by the term $\max_{\tilde{\pi}} p(\tilde{\pi}; \boldsymbol{s})$. Although $p_L(Y; \boldsymbol{\omega}, \boldsymbol{s})$ appears looser than $p_U(Y; \boldsymbol{\omega}, \boldsymbol{s})$ for certain configurations of $\boldsymbol{\omega}$ and $\boldsymbol{s}$, it provides more consistent results across all hyperparameter combinations.

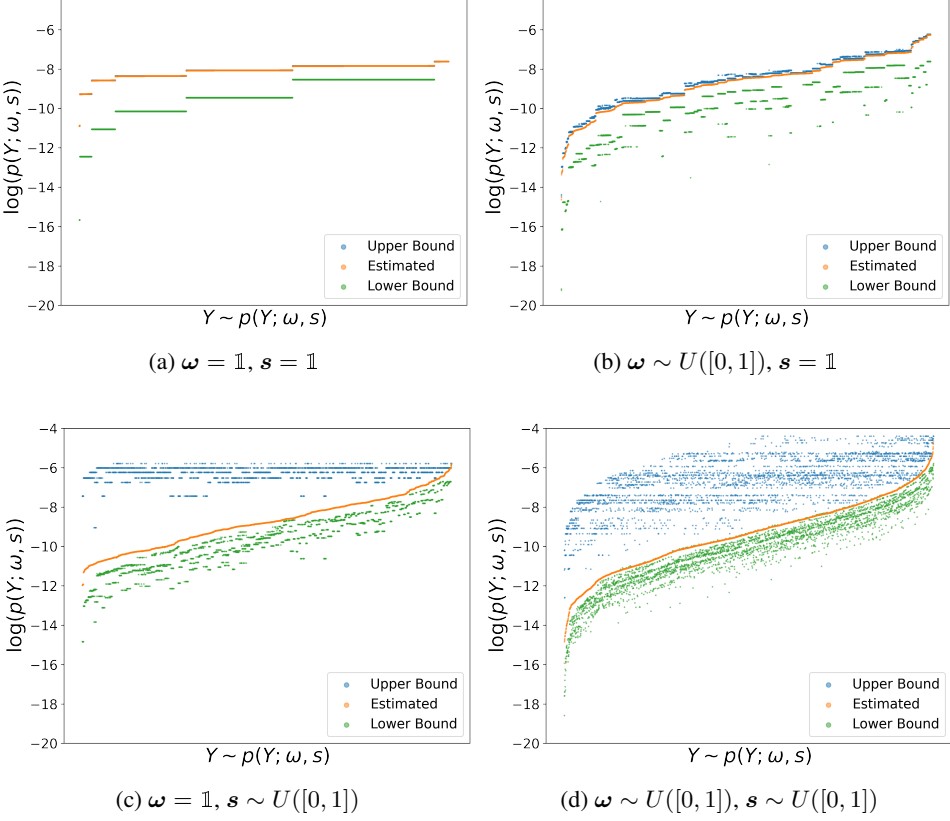

Figure 5: Partitions with $n = 5$ and $K = 5$ for different $\boldsymbol{\omega}$ and $\boldsymbol{s}$. Each point in the plots corresponds to one of the $n^K$ different partitions and their respective estimated probability mass and its upper/lower bounds according to Lemma 4.2.

## C.2 Variational Clustering with Random Partition Models

### C.2.1 Loss Function

As mentioned in Section 5.1, for a given dataset $X$ with $N$ samples, let $Z$ and $Y$ contain the respective latent vectors and cluster assignments for each sample in $X$. The generative process can then be summarized as follows: First, we sample the cluster assignments $Y$ from an RPM, i.e., $Y \sim P(Y; \boldsymbol{\omega}, \boldsymbol{s})$. Given $Y$, we can sample the latent variables $Z$, where for each $\boldsymbol{y}$ we have $\boldsymbol{z} \sim \mathcal{N}(\boldsymbol{\mu_y}, \boldsymbol{\sigma_y^T}\mathbb{I}_l)$, $\boldsymbol{z} \in \mathbb{R}^l$. Finally, we sample $X$ by passing each $\boldsymbol{z}$ through a decoder like in vanilla VAEs. Using Bayes rule and Jensen's inequality, we can then derive the following evidence lower bound (ELBO):

$$
\begin{aligned}
\log(p(X)) &= \log\left(\int \sum_Y p(X, Y, Z)dZ\right) \\
&\geq \mathbb{E}_{q(Z,Y|X)}\left[\log\left(\frac{p(X|Z)p(Z|Y)p(Y)}{q(Z,Y|X)}\right)\right] \\
&:= \mathcal{L}_{ELBO}(X)
\end{aligned}
$$

We then assume that we can factorize the approximate posterior as follows:

$$
q(Z, Y|X) = q(Y|X)\prod_{\boldsymbol{x} \in X} q(\boldsymbol{z}|\boldsymbol{x})
$$

Note that while we do assume conditional independence between $\boldsymbol{z}$ given its corresponding $\boldsymbol{x}$, we model $q(Y|X)$ with the DRPM and do not have to assume conditional independence between

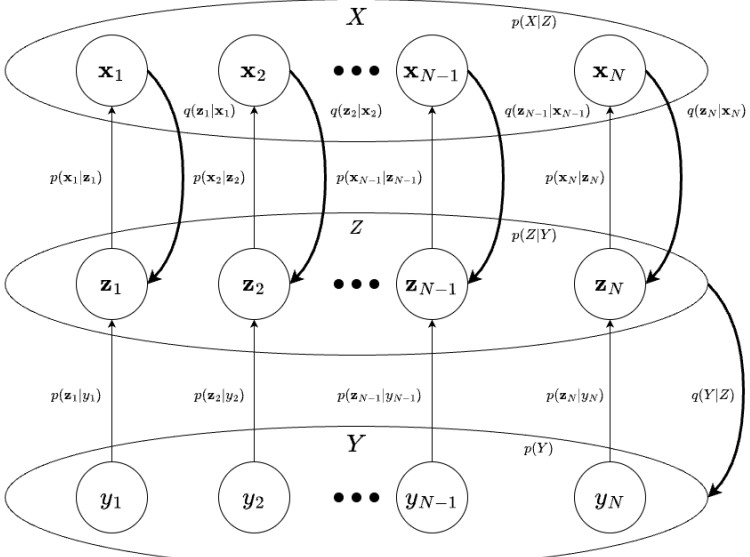

Figure 6: Generative model of the DRPM clustering model. Generative paths are marked with thin arrows, whereas inference is in bold.

different cluster assignments. This allows us to leverage dependencies between samples from the dataset. Hence, we can rewrite the ELBO as follows:

$$
\begin{aligned}
\mathcal{L}_{ELBO}(X) =& \mathbb{E}_{q(Z|X)}\left[\log(p(X|Z))\right] \\
&- \mathbb{E}_{q(Y|X)}\left[KL[q(Z|X)||p(Z|Y)]\right] \\
&- KL[q(Y|X)||p(Y)] \\
=& \sum_{\boldsymbol{x}\in X} \mathbb{E}_{q(\boldsymbol{z}|\boldsymbol{x})}\left[\log p(\boldsymbol{x}|\boldsymbol{z})\right] \\
&- \sum_{\boldsymbol{x}\in X} \mathbb{E}_{q(Y|X)}\left[KL[q(\boldsymbol{z}|\boldsymbol{x})||p(\boldsymbol{z}|Y)]\right] \\
&- KL[q(Y|X)||p(Y)]
\end{aligned}
$$

See Figure 6 for an illustration of the generative process and the assumed inference model. Since computing $P(Y)$ and $q(Y|X)$ is intractable, we further apply Lemma 4.2 to approximate the KL-Divergence term in $\mathcal{L}_{ELBO}$, leading to the following lower bound:

$$
\mathcal{L}_{ELBO} \geq \sum_{\boldsymbol{x}\in X} \mathbb{E}_{q(\boldsymbol{z}|\boldsymbol{x})}\left[\log p(\boldsymbol{x}|\boldsymbol{z})\right] \tag{43}
$$

$$
- \sum_{\boldsymbol{x}\in X} \mathbb{E}_{q(Y|X)}\left[KL[q(\boldsymbol{z}|\boldsymbol{x})||p(\boldsymbol{z}|Y)]\right] \tag{44}
$$

$$
- \mathbb{E}_{q(Y|X)}\left[\log \frac{|\Pi_Y| \cdot q(\boldsymbol{n};\boldsymbol{\omega}(X))}{p(\boldsymbol{n};\boldsymbol{\omega})p(\pi_Y;\boldsymbol{s})}\right] \tag{45}
$$

$$
- \log\left(\max_{\tilde{\pi}} q(\tilde{\pi};\boldsymbol{s}(X))\right), \tag{46}
$$

where $\pi_Y$ is the permutation that lead to $Y$ during the two-stage resampling process. Further, we want to control the regularization strength of the KL divergences similar to the $\beta$-VAE (Higgins et al., 2016). Since the different terms have different regularizing effects, we rewrite Equations (45)

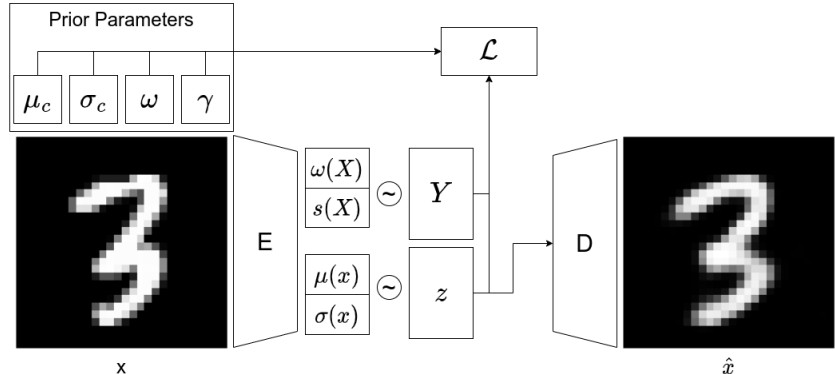

Figure 7: Autoencoder architecture of the DRPM-VC model.

and (46) and weight the individual terms as follows, leading to our final loss:

$$\mathcal{L} := - \sum_{\boldsymbol{x} \in X} \mathbb{E}_{q(\boldsymbol{z}|\boldsymbol{x})} \left[ \log p(\boldsymbol{x}|\boldsymbol{z}) \right] \tag{47}$$

$$+ \beta \cdot \sum_{\boldsymbol{x} \in X} \mathbb{E}_{q(Y|X)} \left[ KL[q(\boldsymbol{z}|\boldsymbol{x})||p(\boldsymbol{z}|Y)] \right] \tag{48}$$

$$+ \gamma \cdot \mathbb{E}_{q(Y|X)} \left[ \log \left( \frac{|\Pi_Y| \cdot q(\boldsymbol{n}; \boldsymbol{\omega}(X))}{p(\boldsymbol{n}; \boldsymbol{\omega})} \right) \right] \tag{49}$$

$$+ \delta \cdot \mathbb{E}_{q(Y|X)} \left[ \log \left( \frac{\max_{\tilde{\pi}} q(\tilde{\pi}; \boldsymbol{s}(X))}{p(\pi_Y; \boldsymbol{s})} \right) \right] \tag{50}$$

### C.2.2   Architecture

The model for our clustering experiments is a relatively simple, fully-connected autoencoder with a structure as seen in Figure 7. We have a fully connected encoder $E$ with three layers mapping the input to $500$, $500$, and $2000$ neurons, respectively. We then compute each parameter by passing the encoder output through a linear layer and mapping to the respective parameter dimension in the last layer. In our experiments, we use a latent dimension size of $l = 10$ for MNIST and $l = 20$ for FMNIST, such that $\boldsymbol{\mu}(\boldsymbol{x}), \boldsymbol{\sigma}(\boldsymbol{x}) \in \mathbb{R}^l$. To understand the architecture choice for the DRPM parameters, let us first take a closer look at Equation (48). For each sample $\boldsymbol{x}$, this term minimizes the expected KL divergence between its approximate posterior $q(\boldsymbol{z}|\boldsymbol{x}) = \mathcal{N}(\boldsymbol{\mu}(\boldsymbol{x}), \mathrm{diag}(\boldsymbol{\sigma}(\boldsymbol{x})))$ and the prior at index $\boldsymbol{y}$ given by the partition $Y$ sampled from the DRPM $q(Y|X; \boldsymbol{s}, \boldsymbol{\omega})$, i.e., $\mathcal{N}(\boldsymbol{\mu_y}, \mathrm{diag}(\boldsymbol{\sigma_y}))$. Ideally, the most likely partition should assign the approximate posterior to the prior that minimizes this KL divergence. We can compute such $\boldsymbol{s}(X)$ and $\boldsymbol{\omega}(X)$ given the parameters of the approximate posterior and priors as follows:

$$\forall \boldsymbol{x}_i \in X : s_i(\boldsymbol{x}_i) = u \cdot \left( K - \arg \min_k \left( KL[\mathcal{N}(\boldsymbol{\mu}(\boldsymbol{x}_i), \mathrm{diag}(\boldsymbol{\sigma}(\boldsymbol{x}_i)))||\mathcal{N}(\boldsymbol{\mu}_k, \mathrm{diag}(\boldsymbol{\sigma}_k))] \right) \right)$$

$$\boldsymbol{\omega}(X) = \frac{1}{|X|} \sum_{\boldsymbol{x} \in X}^N \left\{ \frac{\mathcal{N}(\boldsymbol{x}|\boldsymbol{\mu}_k, \mathrm{diag}(\boldsymbol{\sigma}_k))}{\sum_{k'=1}^K \mathcal{N}(\boldsymbol{x}|\boldsymbol{\mu}_{k'}, \mathrm{diag}(\boldsymbol{\sigma}_{k'}))} \right\}_{k=1}^K,$$

where $u$ is a scaling constant that controls the probability of sampling the most likely partition. Note that $\boldsymbol{\omega}$ and $\boldsymbol{s}$ minimize Equation (48) if defined this way when given the distribution parameters of the approximate posterior and the priors. The only thing that is left unclear is how much $u$ should scale the scores $\boldsymbol{s}$. Ultimately, we leave $u$ as a learnable parameter but detach the rest of the computation of $\boldsymbol{s}$ and $\boldsymbol{\omega}$ from the computational graph to improve stability during training. Finally, once we resample $z \sim \mathcal{N}(\mu(\boldsymbol{x}), \sigma(\boldsymbol{x}))$, we pass it through a fully connected decoder $D$ with four layers mapping $z$ to $2000$, $500$, and $500$ neurons in the first three layers and then finally back to the input dimension in the last layer to end up with the reconstructed sample $\hat{\boldsymbol{x}}$.

### C.2.3 Training

As in vanilla VAEs, we can estimate the reconstruction term in Equation (47) with MCMC by applying the reparametrization trick (Kingma and Welling, 2014) to $q(\boldsymbol{z}|\boldsymbol{x})$ to sample $M$ samples $\boldsymbol{z}^{(i)} \sim q(\boldsymbol{z}|\boldsymbol{x})$ and compute their reconstruction error to estimate Equation (47). Similarly, we can sample from $q(Y|X)$ $L$ times to estimate the terms in Equations (48) to (50), such that we minimize

$$
\begin{aligned}
\tilde{\mathcal{L}} :=& -\sum_{\boldsymbol{x} \in X} \frac{1}{M} \sum_{i=1}^{M} \log p(\boldsymbol{x}|\boldsymbol{z}^{(i)}) \\
&+ \frac{\beta}{L} \cdot \sum_{\boldsymbol{x} \in X} \sum_{i=1}^{L} KL[q(\boldsymbol{z}|\boldsymbol{x})||p(\boldsymbol{z}|Y^{(i)})] \\
&+ \frac{\gamma}{L} \cdot \sum_{i=1}^{L} \log \left( \frac{|\Pi_{Y^{(i)}}| \cdot q(\boldsymbol{n}^{(i)}; \boldsymbol{\omega}(X))}{p(\boldsymbol{n}^{(i)}; \boldsymbol{\omega})} \right) \\
&+ \frac{\delta}{L} \cdot \sum_{i=1}^{L} \log \left( \frac{\max_{\tilde{\pi}} q(\tilde{\pi}; \boldsymbol{s}(X))}{p(\pi_{Y^{(i)}}; \boldsymbol{s})} \right)
\end{aligned}
$$

In our experiments, we set $M = 1$ and $L = 100$ since the MVHG and PL distributions are not concentrated around their mean very well, and more Monte Carlo samples thus lead to better approximations of the expectation terms. We further set $\beta = 1$ for MNIST and $\beta = 0.1$ for FMNIST, and otherwise $\gamma = 1$, and $\delta = 0.01$ for all experiments.

To resample $\boldsymbol{n}$ and $\pi$ we need to apply temperature annealing (Grover et al., 2019; Sutter et al., 2023). To do this, we applied the exponential schedule that was originally proposed together with the Gumbel-Softmax trick (Jang et al., 2016; Maddison et al., 2017), i.e., $\tau = \max(\tau_{final}, exp(-rt))$, where $t$ is the current training step and $r$ is the annealing rate. For our experiments, we choose $r = \frac{\log(\tau_{final}) - \log(\tau_{init})}{100000}$ in order to annealing over 100000 training step. Like Jang et al. (2016), we set $\tau_{init} = 1$ and $\tau_{final} = 0.5$.

Similar to Jiang et al. (2016), we quickly realized that proper initialization of the cluster parameters and network weights is crucial for variational clustering. In our experiments, we pretrained the autoencoder structure by adapting the contrastive loss of (Li et al., 2022), as they demonstrated that their representations manage to retain clusters in low-dimensional space. Further, we also added a reconstruction loss to initialize the decoder properly. To initialize the prior parameters, we fit a GMM to the pretrained embeddings of the training set and took the resulting Gaussian parameters to initialize our priors. Note that we used the same initialization across all baselines. See Appendix C.2.4 for an ablation where we pretrain with only a reconstruction loss similar to what was proposed with the VaDE baseline.

To optimize the DRPM-VC in our experiments, we used the AdamW (Loshchilov and Hutter, 2019) optimizer with a learning rate of 0.0001 with a batch size of 256 for 1024 epochs. During initial experiments with the DRPM-VC, we realized that the pretrained weights of the encoder would often lose the learned structure in the first couple of training epochs. We suspect this to be an artifact of instabilities induced by temperature annealing. To deal with these problems, we decided to freeze the first three layers of the encoder when training the DRPM-VC, giving us much better results. See Appendix C.2.5 for an ablation where we applied the same optimization procedure to VaDE.

Finally, when training the VaDE baseline and the DRPM-VC on FMNIST, we often observe a local optimum where the prior distributions collapse and become identical. We can solve this problem by refitting the GMM in the latent space every 10 epochs and by using the resulting parameters to reinitialize the prior distributions.

### C.2.4 Reconstruction Pretraining

While the results of our variational clustering method depend a lot on the specific pretraining, we want to demonstrate that improvements over the baselines do not depend on the chosen pretraining method. To that end, we repeat our experiments but initialize the weights of our model with an autoencoder that has been trained to minimize the mean squared error between the input and the reconstruction. This initialization procedure was originally proposed in (Jiang et al., 2016). We

Table 4: We compare the clustering performance of the DRPM-VC on test sets of MNIST and FMNIST between GMM in latent space (Latent GMM) and Variational Deep Embedding (VaDE) initializing weights using an autoencoder trained on a reconstruction objective. We measure performance in terms of the Normalized Mutual Information (NMI), Adjusted Rand Index (ARI), and cluster accuracy (ACC) over five seeds and put the best model in bold.

| | MNIST | | | FMNIST | | |
| --- | --- | --- | --- | --- | --- | --- |
| | NMI | ARI | ACC | NMI | ARI | ACC |
| LATENT GMM | $0.75_{\pm 0.00}$ | $0.66_{\pm 0.01}$ | $\mathbf{0.75}_{\pm 0.01}$ | $0.56_{\pm 0.02}$ | $0.41_{\pm 0.03}$ | $0.57_{\pm 0.02}$ |
| VADE | $\mathbf{0.77}_{\pm 0.02}$ | $0.62_{\pm 0.04}$ | $0.69_{\pm 0.04}$ | $0.53_{\pm 0.07}$ | $0.35_{\pm 0.08}$ | $0.47_{\pm 0.09}$ |
| DRPM-VC | $0.74_{\pm 0.00}$ | $\mathbf{0.67}_{\pm 0.01}$ | $\mathbf{0.75}_{\pm 0.02}$ | $\mathbf{0.59}_{\pm 0.01}$ | $\mathbf{0.47}_{\pm 0.02}$ | $\mathbf{0.62}_{\pm 0.01}$ |

Table 5: We compare the clustering performance of the DRPM-VC on test sets of MNIST and FMNIST between GMM in latent space (Latent GMM), and Variational Deep Embedding (VaDE) when freezing the encoder. We measure performance in terms of the Normalized Mutual Information (NMI), Adjusted Rand Index (ARI), and cluster accuracy (ACC) over five seeds and put the best model in bold.

| | MNIST | | | FMNIST | | |
| --- | --- | --- | --- | --- | --- | --- |
| | NMI | ARI | ACC | NMI | ARI | ACC |
| LATENT GMM | $0.86_{\pm 0.02}$ | $0.83_{\pm 0.06}$ | $0.88_{\pm 0.07}$ | $0.60_{\pm 0.00}$ | $0.47_{\pm 0.01}$ | $0.62_{\pm 0.01}$ |
| VADE | $\mathbf{0.90}_{\pm 0.02}$ | $\mathbf{0.88}_{\pm 0.06}$ | $0.92_{\pm 0.06}$ | $\mathbf{0.64}_{\pm 0.01}$ | $0.47_{\pm 0.01}$ | $0.59_{\pm 0.03}$ |
| DRPM-VC | $0.89_{\pm 0.01}$ | $\mathbf{0.88}_{\pm 0.03}$ | $\mathbf{0.94}_{\pm 0.02}$ | $\mathbf{0.64}_{\pm 0.00}$ | $\mathbf{0.51}_{\pm 0.01}$ | $\mathbf{0.65}_{\pm 0.00}$ |

present the results of this ablation in Table 4. Simply minimizing the reconstruction error does not necessarily retain cluster structures in the latent space. Thus, it does not come as a surprise that overall results get about $10\%$ to $20\%$ worse across most metrics, especially for MNIST, while results on FMNIST only slightly decrease. However, we still beat the baselines across most metrics, suggesting that modeling the implicit dependencies between cluster assignments helps to improve variational clustering performance.

### C.2.5 Baselines with fixed Encoder

For the experiments in the main text, we wanted to implement the VaDE baseline similar to the original method proposed in Jiang et al. (2016). This means, in contrast to our method, we used their optimization procedure, i.e., Adam with a learning rate of $0.002$ with a decay of $0.95$ every 10 steps, and did not freeze the encoder as we do for the DRPM-VC. To ensure our results do not stem from this minor discrepancy, we perform an ablation experiment on VaDE using the same optimizer and learning rate as with the DRPM-VC and freeze the encoder backbone. The results of this additional experiment can be found in Appendix C.2.5. As can be seen, VaDE results do improve when adjusting the optimization procedure in this way. However, we still match or improve upon the results of VaDE in most metrics, especially in ARI and ACC, suggesting purer clusters compared to VaDE. We suspect this is because we assign samples to fixed clusters when sampling from the DRPM, whereas VaDE performs soft assignments by marginalizing over a categorical distribution.

### C.2.6 Additional Partition Samples

In Section 5.1, we have seen a sample of a partition of the DRPM-VC trained on FMNIST. We provide additional samples for both MNIST and FMNIST at the end of the appendix in Figures 15 and 16. We can see that for both datasets, the DRPM-VC learns coherent representations of each cluster that easily allow us to generate new samples from each class.

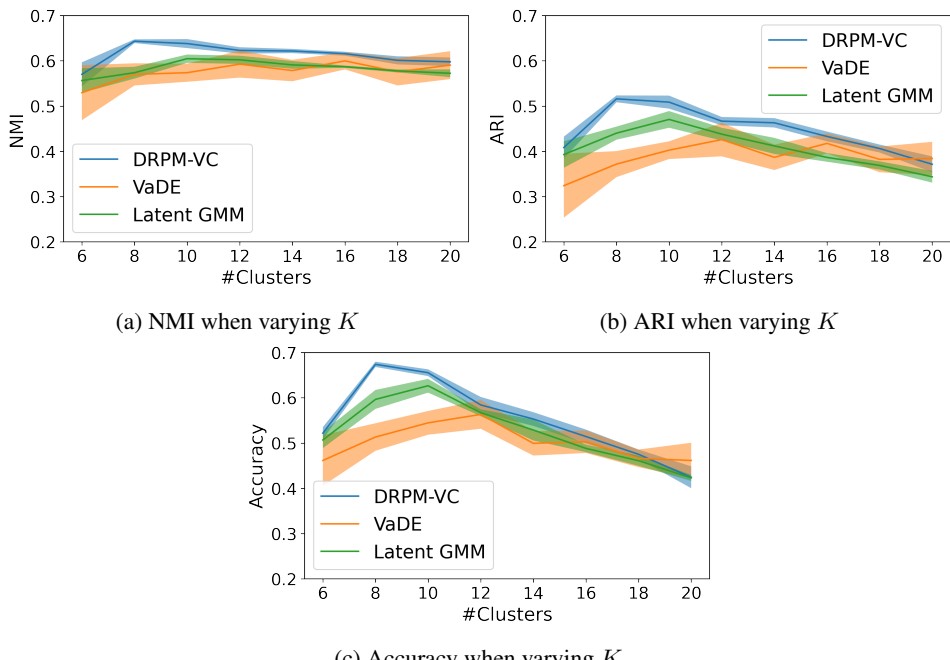

(a) NMI when varying $K$       (b) ARI when varying $K$

(c) Accuracy when varying $K$

Figure 8: Clustering performance of Latent GMM, VaDE, and DRPM-VC when varying the number of clusters $K$ for $5$ seeds on Fashion MNIST. DRPM-VC consistently outperforms the baselines except for $K = 20$, where all methods perform similarly.

### C.2.7 Samples per cluster

In addition to sampling partitions and then generating samples according to the sampled cluster assignments, we can also directly sample from each of the learned priors. We show some examples of this for both MNIST and FMNIST at the end of the appendix in Figures 17 and 18. We can again see that the DRPM-VC learns accurate cluster representations since each of the samples seems to correspond to one of the classes in the datasets. Further, the clusters also seem to capture the diversity in each cluster, as we see a lot of variety across the generated samples.

### C.2.8 Varying the number of Clusters

In previous experiments, we assumed that we had access to the true number of clusters of the dataset, which is, of course, not true in practice. We thus also want to investigate the behavior of the DRPM-VC when varying the number of partitions $K$ of the DRPM and compare it to our baselines. In Figure 8, we show the performance of Latent GMM, VaDE, and DRPM-VC for $K \in \{6, 8, 10, 12, 14, 16, 18, 20\}$ across $5$ different seeds on FMNIST. DRPM-VC clearly outperforms the two baselines for all $K$, except for the extreme case of $K = 20$, where all models seem to perform similarly. Expectedly, DRPM-VC performs well when $K$ is close to the true number of clusters, but performance decreases the farther we are from it. To investigate whether the model still learns meaningful patterns when we are far from the true number of clusters, we additionally generate samples from each prior of the DRPM-VC in Figure 9. Interestingly, we can see that DRPM-VC still detects certain structures in the dataset but starts breaking specific FMNIST categories apart. For instance, it splits the clusters sandals/boots into clusters with (Priors $6/15$) and without (Priors $4/9$) heels or the cluster T-shirt into clothing of lighter (Prior 0) and darker (Prior 3) color. Thus, DRPM-VC allows us to investigate clusters in datasets hierarchically, where low values of $K$ detect more coarse and higher values of $K$ more fine-grained patterns in the data.

### C.2.9 Clustering of STL-10

We include an additional variational clustering ablation on the STL-10 dataset (Coates et al., 2011) that follows the experimental setup of Jiang et al. (VaDE, 2016). As noted by Jiang et al. (2016),

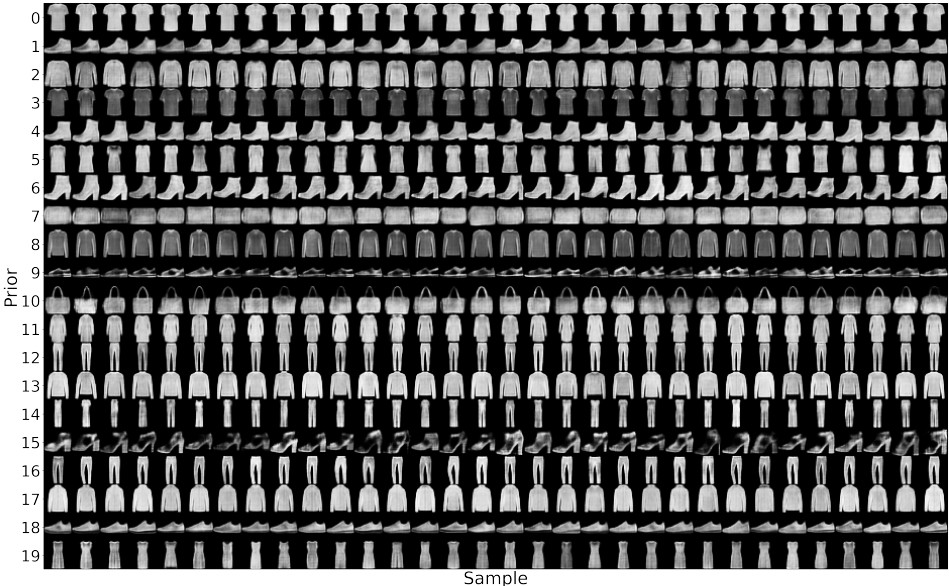

Figure 9: We generate 32 images from each prior when training DRPM-VC on Fashion MNIST with $K = 20$. DRPM-VC starts detecting more fine-grained patterns in the data and splits some of the original clusters, such as sandals, boots, t-shirts, or handbags, into more specific sub-categories.

Table 6: We compare the clustering performance of the DRPM-VC and VaDE on the test set of STL-10 across five seeds. We measure performance in terms of the Normalized Mutual Information (NMI), Adjusted Rand Index (ARI), and cluster accuracy (ACC) over five seeds and put the best model in bold.

|  | STL-10 | | |
| --- | --- | --- | --- |
|  | NMI | ARI | ACC |
| VADE | $0.80_{\pm0.03}$ | $0.76_{\pm0.04}$ | $0.88_{\pm0.02}$ |
| DRPM-VC | $\mathbf{0.83}_{\pm0.01}$ | $\mathbf{0.80}_{\pm0.02}$ | $\mathbf{0.91}_{\pm0.00}$ |

variational clustering algorithms have difficulties clustering in raw pixel space for natural images, which is why we apply DRPM-VC to representations extracted using an Imagenet (Deng et al., 2009) pretrained ResNet-50 (He et al., 2015) as done in Jiang et al. (VaDE, 2016). Note that these results are hard to interpret, as STL-10 is a subset of Imagenet, meaning that representations are relatively easy to cluster as the pretraining task already encourages separation by label. For this reason, we list this experiment separately in the appendix instead of including it in the main text. We present the results of this ablation in Table 6, where we again confirm that modeling cluster assignments with our DRPM can improve upon previous work that modeled the assignments independently.

### C.3 Variational Partitioning of Generative Factors

We assume that we have access to multiple instances or views of the same event, where only a subset of generative factors changes between views. The knowledge about the data collection process provides a form of weak supervision. For example, we have two images of a robot arm as depicted here on the left side (see (Gondal et al., 2019)), which we would describe using high-level concepts such as color, position or rotation degree. From the data collection process, we know that a subset of these generative factors is shared between the two views We do not know how many generative factors there are in total nor how many of them are shared. More precisely, looking at the robot arm, we do not know that the views share two latent factors, depicted in red, out of a total of four factors. Please note that we chose four generative in Figure 10 only for illustrative reason as there are seven generative factors in the *mpi3d* toy dataset. Hence, the goal of learning under weak supervision is not

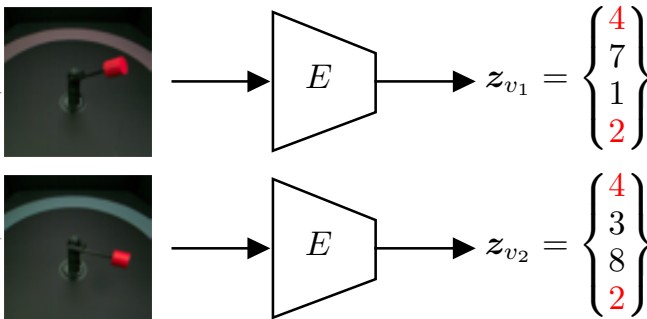

Figure 10: Motivation for the Partitioning of Generative Factors under weak supervision. The knowledge about the data collection process provides a weak supervision signal. We ohave access to a dataset of pairs of images of the same robot arm with a subset of shared generative factors (in red). We want to learn the shared and independent generative factors in addition to learning from the data. The images of the robot arms are taken from Locatello et al. (2020) but originate from the mpi3d toy dataset (see `https://github.com/rr-learning/disentanglement_dataset`). The image is from Sutter et al. (2023) and their ICLR 2023 presentation video (see `https://iclr.cc/virtual/2023/poster/10707`).

only to infer good representations, but also inferring the number of shared and independent generative factors. Learning what is shared and what is independent lets us reason about the group structure without requiring explicit knowledge in the form of expensive labeling. Additionally, leveraging weak supervision and, hence, the underlying group structure holds promise for learning more generalizable and disentangled representations (see (e.g., Locatello et al., 2020)).

### C.3.1 Generative Model

We assume the following generative model for DRPM-VAE

$$p(\boldsymbol{X}) = \int_{\boldsymbol{z}} p(\boldsymbol{X}, \boldsymbol{z}) d\boldsymbol{z} \tag{51}$$

$$= \int_{\boldsymbol{z}} p(\boldsymbol{X} \mid \boldsymbol{z}) p(\boldsymbol{z}) d\boldsymbol{z} \tag{52}$$

where $\boldsymbol{z} = \{\boldsymbol{z}_s, \boldsymbol{z}_1, \boldsymbol{z}_2\}$. The two frames share an unknown number $n_s$ of generative latent factors $\boldsymbol{z}_s$, and an unknown number, $n_1$ and $n_2$, of independent factors $\boldsymbol{z}_1$ and $\boldsymbol{z}_2$. The RPM infers $n_k$ and $\boldsymbol{z}_k$ using $Y$. Hence, the generative model extends to

$$p(\boldsymbol{X}) = \int_{\boldsymbol{z}} p(\boldsymbol{X} \mid \boldsymbol{z}) \sum_Y p(\boldsymbol{z} \mid Y) p(Y) d\boldsymbol{z}$$

$$= \int_{\boldsymbol{z}} p(\boldsymbol{x}_1, \boldsymbol{x}_2 \mid \boldsymbol{z}_s, \boldsymbol{z}_1, \boldsymbol{z}_2) \sum_Y p(\boldsymbol{z} \mid Y) p(Y) d\boldsymbol{z}$$

$$= \int_{\boldsymbol{z}_s, \boldsymbol{z}_1, \boldsymbol{z}_2} p(\boldsymbol{x}_1 \mid \boldsymbol{z}_s, \boldsymbol{z}_1) p(\boldsymbol{x}_2 \mid \boldsymbol{z}_s, \boldsymbol{z}_2) \sum_Y p(\boldsymbol{z}_s, \boldsymbol{z}_1, \boldsymbol{z}_2 \mid Y) p(Y) d\boldsymbol{z}_s d\boldsymbol{z}_1 d\boldsymbol{z}_2 \tag{53}$$

Figure 11 shows the generative and inference models assumptions in a graphical model.

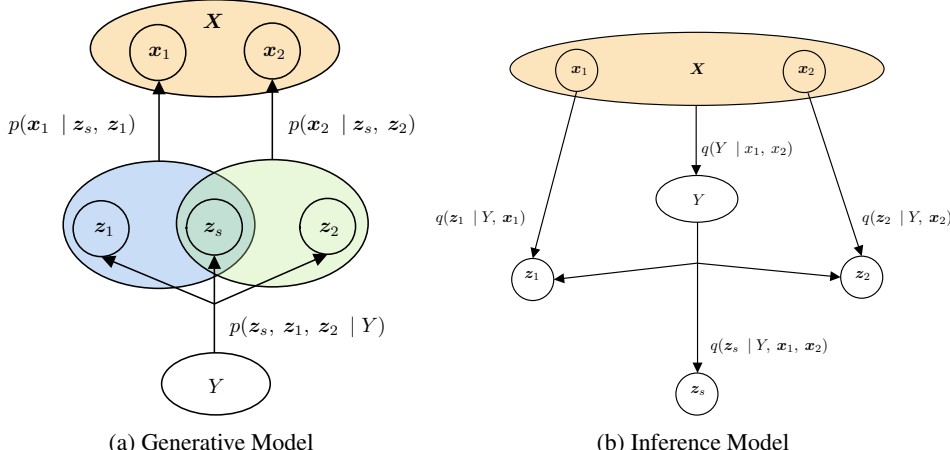

|                    |                    |
|:------------------:|:------------------:|
| (a) Generative Model | (b) Inference Model |

Figure 11: Graphical Models for DRPM-VAE models in the weakly-supervised experiment.

### C.3.2 DRPM ELBO

We derive the following ELBO using the posterior approximation $q(\boldsymbol{z}, Y \mid \boldsymbol{X})$

$$\mathcal{L}_{ELBO}(\boldsymbol{X}) = \mathbb{E}_{q(\boldsymbol{z},Y|\boldsymbol{X})}\left[\log p(\boldsymbol{X} \mid \boldsymbol{z}, Y) - \log \frac{q(\boldsymbol{z}, Y \mid \boldsymbol{X})}{p(\boldsymbol{z}, Y)}\right] \tag{54}$$

$$= \mathbb{E}_{q(\boldsymbol{z},Y|\boldsymbol{X})}\left[\log p(\boldsymbol{X} \mid \boldsymbol{z}) - \log \frac{q(\boldsymbol{z} \mid Y, \boldsymbol{X})q(Y \mid \boldsymbol{X})}{p(\boldsymbol{z})p(Y)}\right] \tag{55}$$

$$= \mathbb{E}_{q(\boldsymbol{z},Y|\boldsymbol{X})}\left[\log p(\boldsymbol{x}_1, \boldsymbol{x}_2 \mid \boldsymbol{z}) - \log \frac{q(\boldsymbol{z} \mid Y, \boldsymbol{X})}{p(\boldsymbol{z})} - \log \frac{q(Y \mid \boldsymbol{X})}{p(Y)}\right] \tag{56}$$

$$= \mathbb{E}_{q(\boldsymbol{z},Y|\boldsymbol{X})}\left[\log p(\boldsymbol{x}_1 \mid \boldsymbol{z}_s, \boldsymbol{z}_1)\right] - \mathbb{E}_{q(\boldsymbol{z},Y|\boldsymbol{X})}\left[\log p(\boldsymbol{x}_2 \mid \boldsymbol{z}_s, \boldsymbol{z}_2)\right]$$
$$- \mathbb{E}_{q(\boldsymbol{z},Y|\boldsymbol{X})}\left[\log \frac{q(\boldsymbol{z}_s, \boldsymbol{z}_1, \boldsymbol{z}_2 \mid Y, \boldsymbol{X})}{p(\boldsymbol{z}_s, \boldsymbol{z}_1, \boldsymbol{z}_2)}\right] - \mathbb{E}_{q(\boldsymbol{z},Y|\boldsymbol{X})}\left[\log \frac{q(Y \mid \boldsymbol{X})}{p(Y)}\right] \tag{57}$$

Following Lemma 4.2, we are able to optimize DRPM-VAE using the following ELBO $\mathcal{L}_{ELBO}(\boldsymbol{X})$:

$$\mathcal{L}_{ELBO} \geq \mathbb{E}_{q(\boldsymbol{z},Y|\boldsymbol{X})}\left[\log p(\boldsymbol{x}_1 \mid \boldsymbol{z}_s, \boldsymbol{z}_1)\right] - \mathbb{E}_{q(\boldsymbol{z},Y|\boldsymbol{X})}\left[\log p(\boldsymbol{x}_2 \mid \boldsymbol{z}_s, \boldsymbol{z}_2)\right] \tag{58}$$

$$- \mathbb{E}_{q(\boldsymbol{z},Y|\boldsymbol{X})}\left[\log \frac{q(\boldsymbol{z}_s, \boldsymbol{z}_1, \boldsymbol{z}_2 \mid Y, \boldsymbol{X})}{p(\boldsymbol{z}_s, \boldsymbol{z}_1, \boldsymbol{z}_2)}\right] \tag{59}$$

$$- \mathbb{E}_{q(Y|\boldsymbol{X})}\left[\log \left(\frac{|\Pi_Y| \cdot q(\boldsymbol{n} \mid \boldsymbol{X}; \boldsymbol{\omega})}{p(\boldsymbol{n}; \boldsymbol{\omega}_p)p(\pi_Y; \boldsymbol{s}_p)}\right)\right] \tag{60}$$

$$- \log \left(\max_{\tilde{\pi}} q(\tilde{\pi} \mid \boldsymbol{X}; \boldsymbol{s})\right), \tag{61}$$

where $\pi_Y$ is the permutation that lead to $Y$ during the two-stage resampling process. Further, we want to control the regularization strength of the KL divergences similar to the $\beta$-VAE (Higgins et al., 2016). The ELBO $\mathcal{L}(\boldsymbol{X})$ to be optimized can be written as

$$\mathcal{L}_{ELBO} = \mathbb{E}_{q(\boldsymbol{z},Y|\boldsymbol{X})}\left[\log p(\boldsymbol{x}_1 \mid \boldsymbol{z}_s, \boldsymbol{z}_1)\right] + \mathbb{E}_{q(\boldsymbol{z},Y|\boldsymbol{X})}\left[\log p(\boldsymbol{x}_2 \mid \boldsymbol{z}_s, \boldsymbol{z}_2)\right] \tag{62}$$

$$- \beta \cdot \mathbb{E}_{q(\boldsymbol{z},Y|\boldsymbol{X})}\left[\log \frac{q(\boldsymbol{z}_s, \boldsymbol{z}_1, \boldsymbol{z}_2 \mid Y, \boldsymbol{X})}{p(\boldsymbol{z}_s, \boldsymbol{z}_1, \boldsymbol{z}_2)}\right] \tag{63}$$

$$- \gamma \cdot \mathbb{E}_{q(Y|\boldsymbol{X})}\left[\log \left(\frac{|\Pi_Y| \cdot q(\boldsymbol{n}; \boldsymbol{\omega}(\boldsymbol{X}))}{p(\boldsymbol{n}; \boldsymbol{\omega}_p)}\right)\right] \tag{64}$$

$$- \delta \cdot \mathbb{E}_{q(Y|\boldsymbol{X})}\left[\log \left(\frac{\max_{\tilde{\pi}} q(\tilde{\pi}; \boldsymbol{s}(\boldsymbol{X}))}{p(\pi_Y; \boldsymbol{s}_p)}\right)\right] \tag{65}$$

where $\boldsymbol{s}(\boldsymbol{X})$ and $\boldsymbol{\omega}(\boldsymbol{X})$ denote distribution parameters, which are inferred from $\boldsymbol{X}$ (similar to the Gaussian parameters in the vanilla VAE).

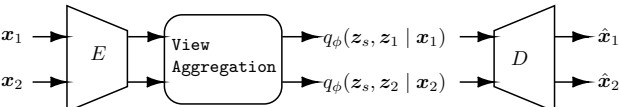

Figure 12: Setup for the weakly-supervised experiment. The three methods differ only in the `View Aggregation` module.

As in vanilla VAEs, we can estimate the reconstruction term in Equation (58) with MCMC by applying the reparametrization trick (Kingma and Welling, 2014) to $q(\boldsymbol{z} \mid Y, \boldsymbol{X})$ to sample $L$ samples $\boldsymbol{z}^{(l)} \sim q(\boldsymbol{z} \mid Y, \boldsymbol{X})$ and compute their reconstruction error to estimate Equation (58). Similarly, we can sample from $q(Y \mid \boldsymbol{X})$ $L$ times. We use $L = 1$ to estimate all expectations in $\mathcal{L}_{ELBO}$.

### C.3.3 Implementation and Hyperparameters

In this experiment, we use the `disentanglement_lib` from Locatello et al. (2020). We use the same architectures proposed in the original paper for all methods we compare to. The baseline algorithms, LabelVAE (Bouchacourt et al., 2018; Hosoya, 2018) and AdaVAE (Locatello et al., 2020) are already implemented in `disentanglement_lib`. For details on the implementation of these methods we refer to the original paper from Locatello et al. (2020). HGVAE is implemented in Sutter et al. (2023). We did not change any hyperparameters or network details. All experiments were performed using $\beta = 1$ as this is the best performing $\beta$ (according to Locatello et al. (2020). For DRPMVAE we chose $\gamma = 0.25$ for all runs. All models are trained on 5 different random seeds and the reported results are averaged over the 5 seeds. We report mean performance with standard deviations.

We adapted Figure 12 from Sutter et al. (2023). It shows the baseline architecture, which is used for all methods. As already stated in the main part of the paper, the methods only differ in the `View Aggregation` module, which determines the shared and independent latent factors. Given a subset $S$ of shared latent factors, we have

$$q_\phi(z_i \mid \boldsymbol{x}_j) = avg(q_\phi(z_i \mid \boldsymbol{x}_1), q_\phi(z_i \mid \boldsymbol{x}_2)) \qquad \forall \ i \in S \qquad (66)$$
$$q_\phi(z_i \mid \boldsymbol{x}_j) = q_\phi(z_i \mid \boldsymbol{x}_j) \qquad \text{else} \qquad (67)$$

where $avg$ is the averaging function of choice (Locatello et al., 2020; Sutter et al., 2023) and $j \in \{1, 2\}$. The methods used (i. e. Label-VAE, Ada-VAE, HG-VAE, DRPM-VAE) differ in how to select the subset S.

For DRPM-VAE, we infer $\boldsymbol{\omega}$ from the pairwise KL-divergences $KL_{pw}$ between the latent vectors of the two views.

$$KL_{pw}(\boldsymbol{x}_1, \boldsymbol{x}_2) = \frac{1}{2} KL[q(\boldsymbol{z}_1 \mid \boldsymbol{x}_1)||q(\boldsymbol{z}_2 \mid \boldsymbol{x}_2)] + \frac{1}{2} KL[q(\boldsymbol{z}_2 \mid \boldsymbol{x}_2)||q(\boldsymbol{z}_1 \mid \boldsymbol{x}_1)] \qquad (68)$$

where $q(\boldsymbol{z}_j \mid \boldsymbol{x}_j)$ are the encoder outputs of the respective images. We do not average or sum across dimensions in the computation of $KL_{pw}(\cdot)$ such that the $KL_{pw}(\cdot)$ is $d$-dimensional, where $d$ is the latent space size. The encoder $E$ in Figure 12 maps to $\boldsymbol{\mu}(\boldsymbol{x}_j)$ and $\boldsymbol{\sigma}(\boldsymbol{x}_j)$ of a Gaussian distribution. Hence, we can compute the KL divergences above in closed form. Afterwards, we feed the pairwise KL divergence $KL_{pw}$ to a single fully-connected layer, which maps from $d$ to $K$ values

$$\log \boldsymbol{\omega} = FC(KL_{pw}(\boldsymbol{x}_1, \boldsymbol{x}_2)) \qquad (69)$$

where $d = 10$ and $K = 2$ in this experiment. $d$ is the total number of latent dimensions and $K$ is the number of groups in the latent space. To infer the scores $\boldsymbol{s}(\boldsymbol{X})$ we again rely on the pairwise KL divergence $KL_{pw}$. Instead of using another fully-connected layer, we directly use the log-values of the pairwise KL divergence

$$\log \boldsymbol{s} = \log KL_{pw}(\boldsymbol{x}_1, \boldsymbol{x}_2) \qquad (70)$$

Similar to the original works, we also anneal the temperature parameter for $p(\boldsymbol{n}; \boldsymbol{\omega})$ and $p(\pi; \boldsymbol{s})$ (Grover et al., 2019; Sutter et al., 2023). We use the same annealing function as in the clustering experiment (see Appendix C.2). We anneal the temperature $\tau$ from 1.0 to 0.5 over the complete training time.

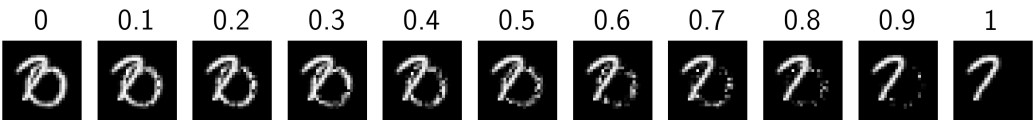

Figure 13: Samples from the noisyMultiMNIST dataset with increasing noise ratio in the right task.

## C.4 Multitask Learning

### C.4.1 MultiMNIST Dataset

The different tasks in multitask learning often vary in difficulty. To measure the effect of discrepancies in task difficulties on DRPM-MTL, we introduce the noisyMultiMNIST dataset.

The noisyMultiMNIST dataset modifies the MultiMNIST dataset (Sabour et al., 2017) as follows. In the right image, we set each pixel value to zero with probability $\alpha \in [0, 1]$. This is done before merging the left and right image in order to only affect the difficulty of the right task. Note that for $\alpha = 0$ noisyMultiMNIST is equivalent to MultiMNIST and for $\alpha = 1$ the right task can no longer be solved. This allows us to control the difficulty of the right task, without changing the difficulty of the left. A few examples are shown in Figure 13.

### C.4.2 Implementation & Architecture

The multitask loss function for the *MultiMNIST* dataset is

$$\mathbb{L} = w_L \mathbb{L}_L + w_R \mathbb{L}_R \tag{71}$$

where $w_L$ and $w_L$ are the loss weights, and $\mathbb{L}_L$ and $\mathbb{L}_R$ are the individual loss terms for the respective tasks $L$ and $R$. In our experiments, we set the task weights to be equal for all dataset versions, i.e. $w_L = w_R = 0.5$. We use these loss weights for the DRPM-MTL and ULS method. For the ULS method, it is by definition and to see the influence of a mismatch in loss weights. The DRPM-MTL method on the other hand does not need additional weighting of loss terms. The task losses are defined as cross-entropy losses

$$\mathbb{L}_t = -\sum_{c=1}^{C_t} \boldsymbol{gt}_c \log \boldsymbol{p}_c = -\boldsymbol{gt}^T \log \boldsymbol{p} \tag{72}$$

where $C_L = C_R = 10$ for MultiMNIST, $\boldsymbol{gt}$ is a one-hot encoded label vector and $\boldsymbol{p}$ is a categorical vector of estimated class assignments probabilities, i.e. $\sum_c \boldsymbol{p}_c = 1$.

The predictions for the individual tasks $\boldsymbol{p}_t$ are given as

$$\boldsymbol{p}_t = h_{\theta_t}(\boldsymbol{z}), \quad \text{where} \tag{73}$$

$$\boldsymbol{z} = \text{enc}_\theta(\boldsymbol{x}) \tag{74}$$

for a sample $\boldsymbol{x} \in \boldsymbol{X}$ (see also Figure 14). We use an adaptation of the LeNet-5 architecture LeCun et al. (1998) to the multitask learning problem (Sener and Koltun, 2018). Both DRPM-MTL and ULS use the same network $\text{enc}_\theta(\cdot)$ with shared architecture up to some layer for both tasks, after which the network branches into two task-specific sub-networks that perform the classifications. Different to the ULS method, the task-specific networks in the DRPM-MTL pipeline predict the digit using only a subset of $\boldsymbol{z}$. DRPM-MTL uses the following prediction scheme

$$\boldsymbol{p}_t = h_{\theta_t}(\boldsymbol{z}_t), \quad \text{where} \tag{75}$$

$$\boldsymbol{z}_t = \boldsymbol{z} \odot \boldsymbol{y}_t \tag{76}$$

$$\boldsymbol{y}_t = \text{DRPM}(\boldsymbol{\omega}, \boldsymbol{s})_t = \text{DRPM}(\text{enc}_\varphi(\boldsymbol{x}))_t \tag{77}$$

The DRPM-MTL encoder first predicts a latent representation $\boldsymbol{z} \leftarrow \text{enc}_\theta(\boldsymbol{x})$, where $\boldsymbol{x}$ is the input image. Using the same encoder architecture but different parameters $\varphi$, we predict a partitioning encoding $\boldsymbol{z}' \leftarrow \text{enc}_\varphi(\boldsymbol{x})$. With a single linear layer per DRPM log-parameter $\log \boldsymbol{\omega}$ and $\log \boldsymbol{s}$ are computed. Next we infer the partition masks $\boldsymbol{y}_L, \boldsymbol{y}_R \sim p(\boldsymbol{y}_L, \boldsymbol{y}_R; \boldsymbol{\omega}, \boldsymbol{s})$. We then feed the masked latent representations $\boldsymbol{z}_L \leftarrow \boldsymbol{z} \odot \boldsymbol{y}_L$ and $\boldsymbol{z}_R \leftarrow \boldsymbol{z} \odot \boldsymbol{y}_R$ into the task specific classification networks $h_{\theta_L}(\boldsymbol{z}_L)$ and $h_{\theta_R}(\boldsymbol{z}_R)$ respectively to obtain the task specific predictions. Since the two tasks in the MultiMNIST dataset are of similar nature, the task-specifc networks $h_{\theta_L}$ and $h_{\theta_R}$ share the same architecture, but have different parameters.

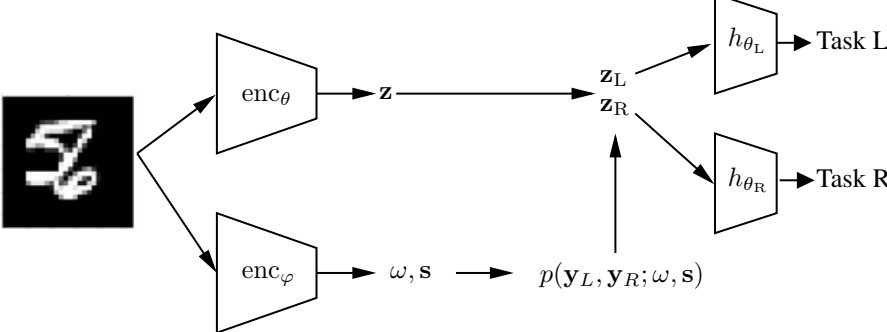

Figure 14: Overview of the multitask learning pipeline of the DRPM-MTL method.

### C.4.3 Training

For both the ULS and the DRPM-MTL model, we use the Adam optimizer with learning rate 0.0005 and train them for 200 epochs with a batch size of 256. We again choose an exponential schedule for the temperature $\tau$ and anneal it over the training time, as is explained in Appendix C.2.3.

In our ablation we use $\alpha \in \{0, 0.1, 0.2, \ldots, 0.9\}$ and train each model with five different seeds. The reported accuracies and partition sizes are then means over the five seeds with the error bands indicating the variance and standard deviation respectively. We evaluate each model after the epoch with the best average test accuracy.

### C.4.4 CelebA for MTL

In addition to the experiment shown in Section 5.3, we show additional results for DRPM-MTL on the CelebA dataset (Liu et al., 2015). In MTL, each of the 40 attributes of the CelebA dataset serves as an individual task. Hence, using CelebA for MTL results is a 40 task learning problem making the scaling of different task losses more difficult compared to MultiMNIST (see Section 5.3) where we only need to scale two different tasks.

We again use the newly introduced DRPM-MTL method and compare it to the ULS model. We use the same pipeline as for MultiMNIST dataset but with different encoders and hyperparameters (see Appendices C.4.2 and C.4.3). We use the pipeline of Sener and Koltun (2018) with a ResNet-based encoder to map an image to a representation of $d = 64$ dimensions. For architectural details, we refer to Sener and Koltun (2018) and `https://github.com/isl-org/MultiObjectiveOptimization`.

Again, ULS inputs all $d = 64$ dimensions to the task-specific sub-networks whereas DRPM-MTL partitions the intermediate representations into $n_T$ different subsets, which are then fed to the respective task networks. $n_T$ is the number of tasks.

Compared to the MultiMNIST experiment (see Appendix C.4.2), we introduce an additional regularization for the DRPM-MTL method. The additional regularization is based on the upper bound in Lemma 4.2 and is penalizing size of $|\Pi_Y|$ for a given $\boldsymbol{n}$. Hence, the loss function changes to

$$\mathbb{L} = \frac{1}{n_T} \sum_{t=1}^{n_T} \mathbb{L}_t + \lambda \cdot \mathbb{L}_{\text{reg}} \tag{78}$$

$$\text{where} \quad \mathbb{L}_{\text{reg}} = \log \left( \prod_{t=1}^{n_T} n_t! \right) = \sum_{t=1}^{n_T} \log \Gamma(n_t + 1) \tag{79}$$

For all versions of the experiment (i.e. $n_T \in 10, 20, 40$), we set $\lambda = 0.015 \approx \frac{1}{64}$, which is the number of elements we want to partition. The task losses $\mathbb{L}_t$ are simple BCE losses similar to the MultiMNIST experiments but with two classes per task only.

We perform two different experiments based on the CelebA experiment. First, we use form a MTL experiments using the first 10 attributes out of the 40 attributes. Second, we increase the number of different tasks to 20. Because we sort the attributes alphabetically in both cases, the first 10 tasks are

Table 7: Results for the MTL experiment on the CelebA dataset. We compare the DRPM-MTL again to the ULS method. We assess the performance of both methods on two sub-experiment of the CelebA experiment. In Table 7a, we form a MTL experiment with 10 different tasks. In Table 7b, we form a MTL experiment with 20 different tasks where the first 10 tasks are the same as in the 10 tasks experiment. And in Table 6c, a MTL experiment with all 40 tasks from the dataset. We train both methods for 50 epochs using a learning rate of 0.0001 and a batch size of 128. The temperature annealing schedule remains the same as in the MultiMNIST experiment. We report the per task classification accuracy in percentages (%) as well as the average task accuracy in the bottow row of both subtables.

(a) 10 Tasks

|  | ULS | DRPM |
|---|---|---|
| T0 | $92.0_{\pm 0.5}$ | $\mathbf{92.4_{\pm 0.5}}$ |
| T1 | $\mathbf{83.8_{\pm 0.4}}$ | $83.7_{\pm 0.2}$ |
| T2 | $80.2_{\pm 0.5}$ | $80.2_{\pm 0.4}$ |
| T3 | $81.9_{\pm 0.8}$ | $\mathbf{82.2_{\pm 0.6}}$ |
| T4 | $98.5_{\pm 0.2}$ | $98.5_{\pm 0.1}$ |
| T5 | $95.2_{\pm 0.2}$ | $\mathbf{95.3_{\pm 0.2}}$ |
| T6 | $80.0_{\pm 1.4}$ | $\mathbf{82.4_{\pm 0.4}}$ |
| T7 | $82.0_{\pm 0.3}$ | $\mathbf{82.2_{\pm 0.2}}$ |
| T8 | $89.7_{\pm 0.7}$ | $\mathbf{90.7_{\pm 0.2}}$ |
| T9 | $94.6_{\pm 0.5}$ | $\mathbf{95.0_{\pm 0.2}}$ |
| avg(Tasks) | $87.8_{\pm 0.3}$ | $\mathbf{88.3_{\pm 0.1}}$ |

(b) 20 Tasks

|  | ULS | DRPM |
|---|---|---|
| T0 | $92.4_{\pm 0.7}$ | $\mathbf{93.0_{\pm 0.2}}$ |
| T1 | $83.7_{\pm 0.6}$ | $\mathbf{83.9_{\pm 0.7}}$ |
| T2 | $79.9_{\pm 0.6}$ | $\mathbf{80.1_{\pm 0.4}}$ |
| T3 | $82.4_{\pm 0.5}$ | $\mathbf{83.0_{\pm 0.7}}$ |
| T4 | $98.6_{\pm 0.1}$ | $98.6_{\pm 0.1}$ |
| T5 | $95.2_{\pm 0.1}$ | $\mathbf{95.5_{\pm 0.0}}$ |
| T6 | $82.0_{\pm 1.3}$ | $\mathbf{84.4_{\pm 0.4}}$ |
| T7 | $82.5_{\pm 0.1}$ | $\mathbf{82.8_{\pm 0.2}}$ |
| T8 | $\mathbf{90.1_{\pm 0.9}}$ | $91.0_{\pm 0.4}$ |
| T9 | $94.7_{\pm 0.2}$ | $\mathbf{95.1_{\pm 0.1}}$ |
| T10 | $95.9_{\pm 0.1}$ | $95.9_{\pm 0.1}$ |
| T11 | $\mathbf{84.9_{\pm 0.1}}$ | $84.6_{\pm 0.3}$ |
| T12 | $91.0_{\pm 0.4}$ | $\mathbf{91.6_{\pm 0.2}}$ |
| T13 | $94.7_{\pm 0.1}$ | $\mathbf{94.9_{\pm 0.1}}$ |
| T14 | $95.4_{\pm 0.3}$ | $\mathbf{96.0_{\pm 0.1}}$ |
| T15 | $99.2_{\pm 0.0}$ | $99.2_{\pm 0.1}$ |
| T16 | $95.8_{\pm 0.3}$ | $\mathbf{96.0_{\pm 0.1}}$ |
| T17 | $97.3_{\pm 0.3}$ | $\mathbf{97.5_{\pm 0.2}}$ |
| T18 | $91.2_{\pm 0.3}$ | $91.2_{\pm 0.1}$ |
| T19 | $87.0_{\pm 0.3}$ | $\mathbf{87.3_{\pm 0.2}}$ |
| avg(Tasks) | $90.7_{\pm 0.2}$ | $\mathbf{91.1_{\pm 0.1}}$ |

shared between the two experiment versions. And third, we set $n_T = 40$, where the first 20 tasks are the shared with the previous experiment.

Table 7 shows the results of all CelebA experiments for both methods, ULS and DRPM-MTL. We see that the DRPM-MTL scales better to a larger number of tasks compared to the ULS method, highlighting the importance of finding new ways of automatic scaling between tasks. Interestingly, the DRPM-MTL outperforms the ULS method on most tasks for the 20-tasks experiment even though it has only access to $d/n_T = 64/20 = 3.2$ dimensions on average. And even more extrem for $n_T = 40$, DRPM-MTL on average has only access to $d/n_T = 64/40 = 1.6$ dimensions. On the other hand, the ULS method can access the full set of 64 dimensions for every single task.

## C.5 Supervised Learning

Given the true partition $Y$, we can also adapt the DRPM to learn partitions in a supervised fashion. One instance where we know the true partition of elements is in the case of classification. There, for a given batch $X$ containing $B$ samples, we have $Y := (\boldsymbol{y}_1, \ldots, \boldsymbol{y}_B)$ where $\boldsymbol{y}_i \in \mathbb{R}^K$ is the one-hot encoding of the label of the $i$-th sample in the batch. In this ablation, we infer $\hat{Y} := DRPM(\boldsymbol{\omega}_{\theta_1}(X), \boldsymbol{s}_{\theta_2}(X))$, where we compute $\boldsymbol{\omega}(X)$ and $\boldsymbol{s}(X)$ as in the variational clustering experiment (Appendix C.2.2) and use the DRPM without resampling as in the multitask experiment (Section 5.3). We optimize

(c) 40 Tasks

|           | ULS           | DRPM          |
|-----------|---------------|---------------|
| T0        | $92.9_{\pm0.6}$ | $92.6_{\pm0.5}$ |
| T1        | $83.4_{\pm0.5}$ | $83.8_{\pm0.5}$ |
| T2        | $80.6_{\pm0.8}$ | $80.6_{\pm0.3}$ |
| T3        | $82.8_{\pm1.1}$ | $83.1_{\pm0.4}$ |
| T4        | $98.6_{\pm0.1}$ | $98.7_{\pm0.1}$ |
| T5        | $95.4_{\pm0.1}$ | $95.4_{\pm0.2}$ |
| T6        | $81.6_{\pm2.2}$ | $84.4_{\pm1.2}$ |
| T7        | $82.7_{\pm0.3}$ | $82.6_{\pm0.2}$ |
| T8        | $90.1_{\pm0.8}$ | $90.7_{\pm0.5}$ |
| T9        | $94.8_{\pm0.2}$ | $95.0_{\pm0.1}$ |
| T10       | $96.0_{\pm0.2}$ | $95.9_{\pm0.1}$ |
| T11       | $85.0_{\pm0.5}$ | $85.0_{\pm0.3}$ |
| T12       | $91.6_{\pm0.6}$ | $92.2_{\pm0.1}$ |
| T13       | $94.8_{\pm0.2}$ | $94.8_{\pm0.2}$ |
| T14       | $95.6_{\pm0.4}$ | $95.7_{\pm0.2}$ |
| T15       | $99.3_{\pm0.1}$ | $99.3_{\pm0.1}$ |
| T16       | $96.2_{\pm0.2}$ | $96.1_{\pm0.1}$ |
| T17       | $97.4_{\pm0.2}$ | $97.5_{\pm0.1}$ |
| T18       | $91.1_{\pm0.1}$ | $91.3_{\pm0.3}$ |
| T19       | $87.1_{\pm0.3}$ | $87.0_{\pm0.5}$ |
| T20       | $98.6_{\pm0.0}$ | $98.5_{\pm0.1}$ |
| T21       | $93.6_{\pm0.1}$ | $93.6_{\pm0.2}$ |
| T22       | $96.0_{\pm0.1}$ | $95.9_{\pm0.2}$ |
| T23       | $91.8_{\pm0.4}$ | $92.6_{\pm0.6}$ |
| T24       | $95.4_{\pm0.2}$ | $95.5_{\pm0.1}$ |
| T25       | $71.5_{\pm1.6}$ | $72.9_{\pm1.2}$ |
| T26       | $96.0_{\pm0.4}$ | $96.4_{\pm0.2}$ |
| T27       | $74.9_{\pm0.5}$ | $76.1_{\pm0.3}$ |
| T28       | $93.5_{\pm0.5}$ | $94.0_{\pm0.3}$ |
| T29       | $92.8_{\pm0.2}$ | $93.5_{\pm0.2}$ |
| T30       | $96.3_{\pm0.3}$ | $96.1_{\pm0.3}$ |
| T31       | $92.7_{\pm0.2}$ | $93.0_{\pm0.1}$ |
| T32       | $81.0_{\pm0.4}$ | $82.1_{\pm0.4}$ |
| T33       | $83.3_{\pm0.2}$ | $83.8_{\pm0.6}$ |
| T34       | $89.2_{\pm0.2}$ | $89.3_{\pm0.1}$ |
| T35       | $98.9_{\pm0.0}$ | $99.0_{\pm0.0}$ |
| T36       | $91.7_{\pm0.1}$ | $91.8_{\pm0.2}$ |
| T37       | $87.6_{\pm0.6}$ | $88.1_{\pm0.1}$ |
| T38       | $95.5_{\pm0.5}$ | $95.6_{\pm0.2}$ |
| T39       | $87.9_{\pm0.1}$ | $82.7_{\pm3.2}$ |
| avg(Tasks) | $90.6_{\pm0.1}$ | $90.8_{\pm0.1}$ |

parameters $\theta_1$ and $\theta_2$ by minimizing the following loss:

$$\mathcal{L}(X,Y) := \mathcal{L}_1(X,Y) + \alpha \mathcal{L}_2(X,Y)$$

$$\mathcal{L}_1(X,Y) := \frac{1}{B}\mathcal{L}_{CE}(\hat{Y},Y)$$

$$\mathcal{L}_2(X,Y) := \frac{1}{K}\|\boldsymbol{n}(X) - \sum_{i=1}^{B}\boldsymbol{y}_i\|^2,$$

where $\mathcal{L}_{CE}$ denotes the standard cross-entropy loss, $\boldsymbol{n}$ denotes the output of the MVHG leading to $\hat{Y}$, and $\mathcal{L}_2$ ensures that $\boldsymbol{n}_i$ matches the number of appearances of label $i$ in the current batch. Using this simple training scheme, we achieve an f1-score of $96.43 \pm 0.02$ and $82.71 \pm 0.03$ on MNIST and FMNIST, respectively, further demonstrating the applicability and versatility of the DRPM to a number of different problems.

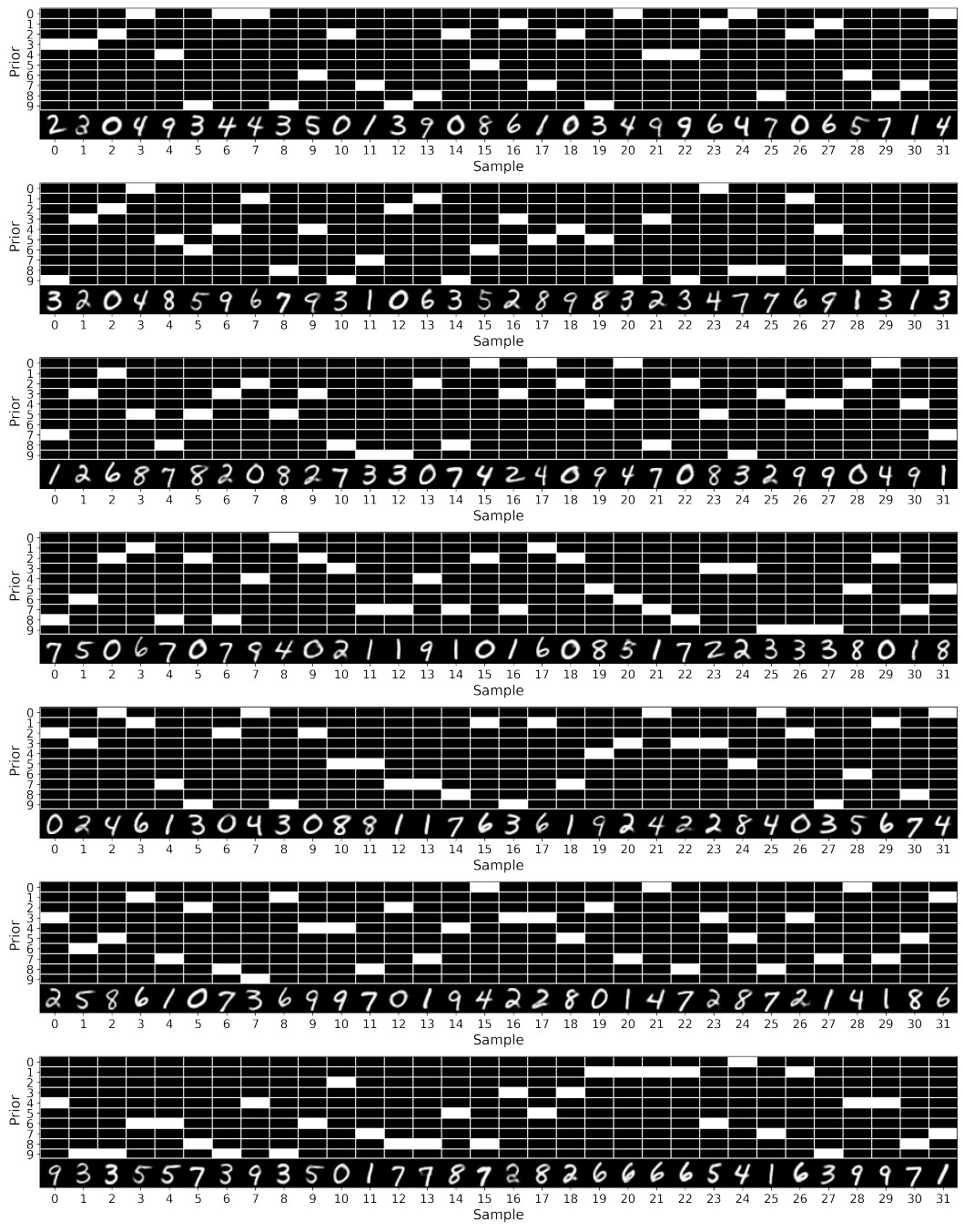

Figure 15: Additional partition samples from the DRPM-VC trained on MNIST. The different sets of each partition match each of the digits very well, even after repeatedly sampling from the model.

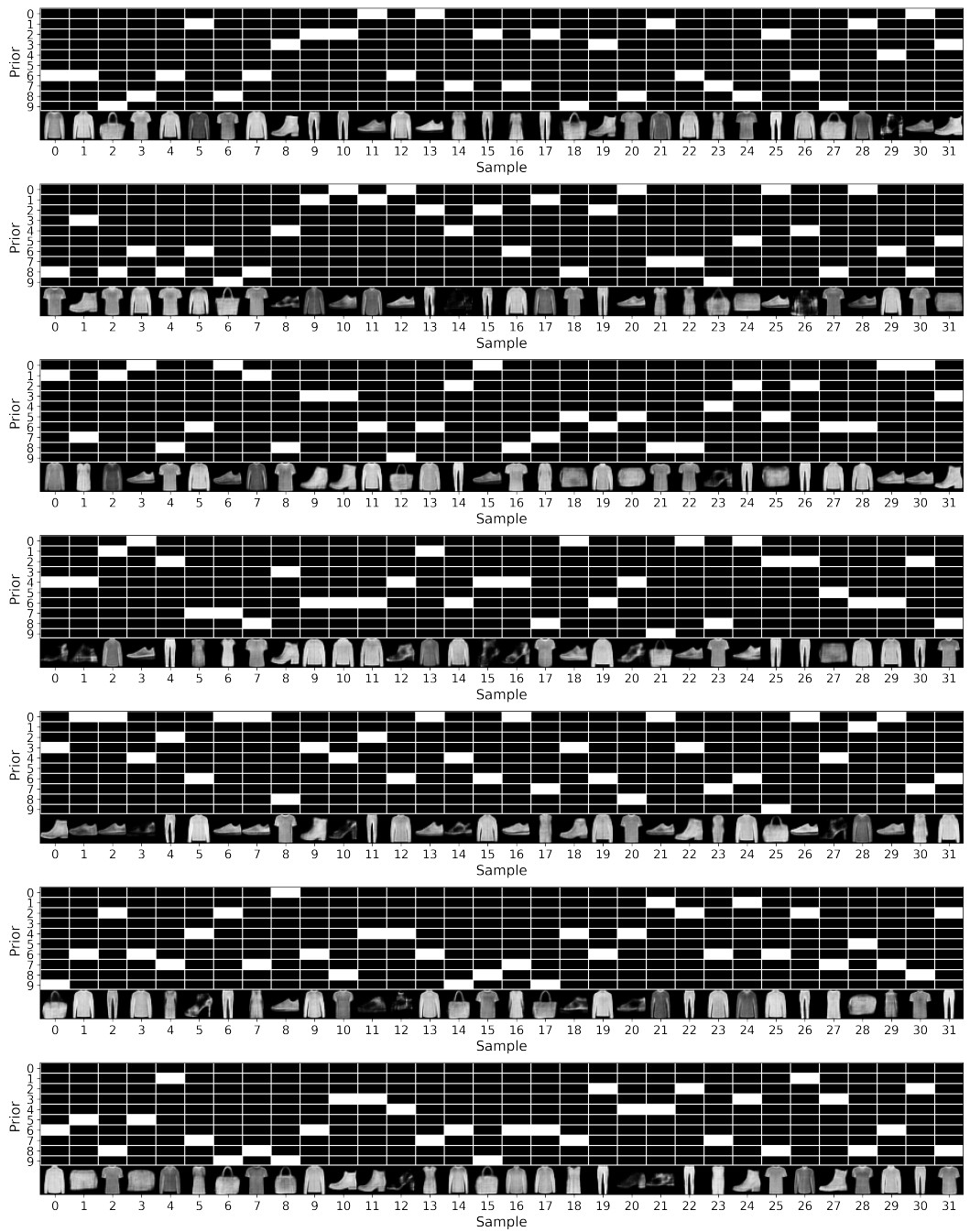

Figure 16: Additional partition samples from the DRPM-VC trained on FMNIST. Most clusters accurately represent one of the clothing categories and generate new samples very well. The only problem is with the handbag class, where the DRPM-VC learns two different clusters for different kinds of handbags (cluster 5 and 6).

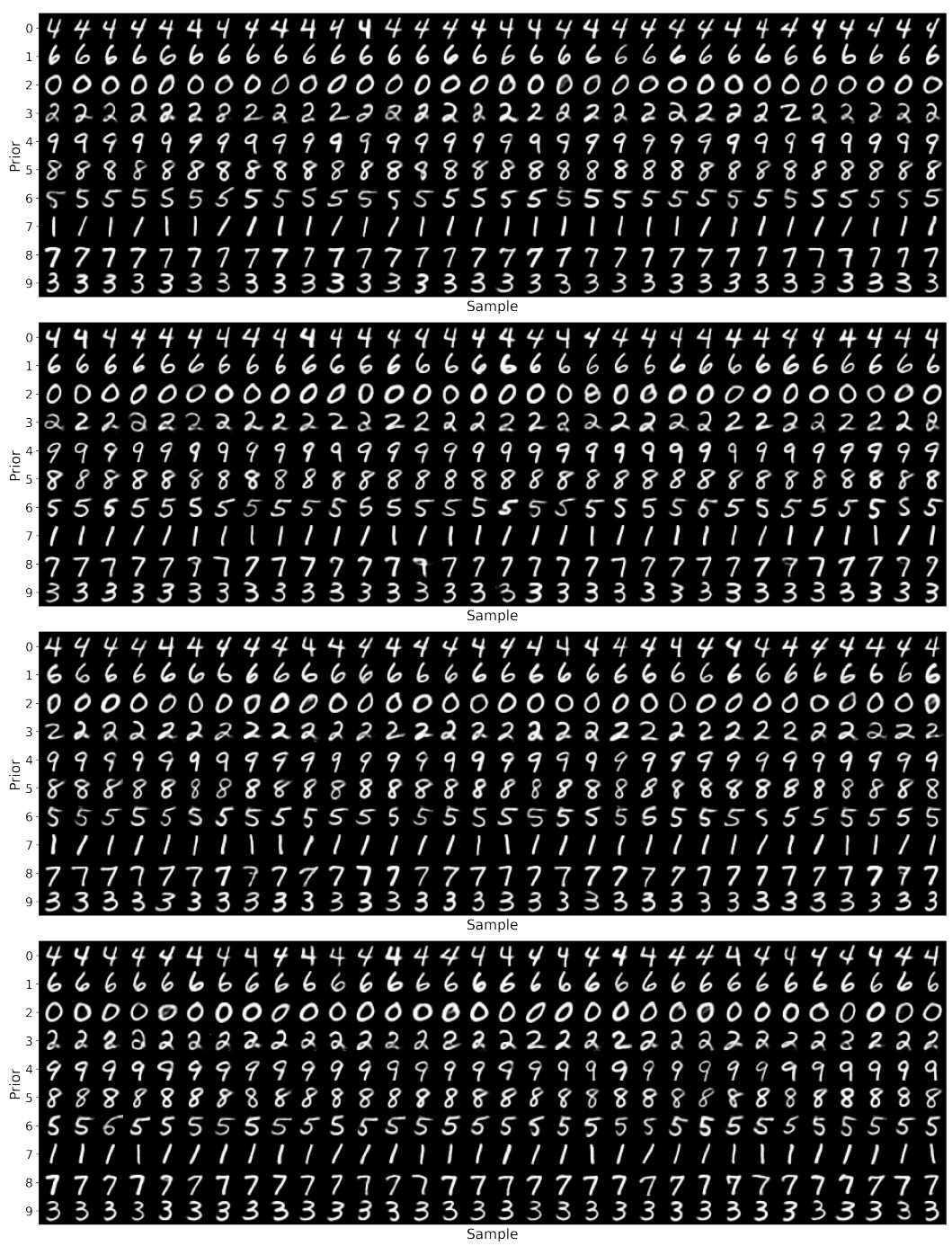

Figure 17: Various samples from each of the generative priors. Each prior learns to represent one of the digits. Further, we see a lot of variation between the different samples, suggesting that the clusters of the DRPM-VC manage to capture some of the diversity present in the dataset.

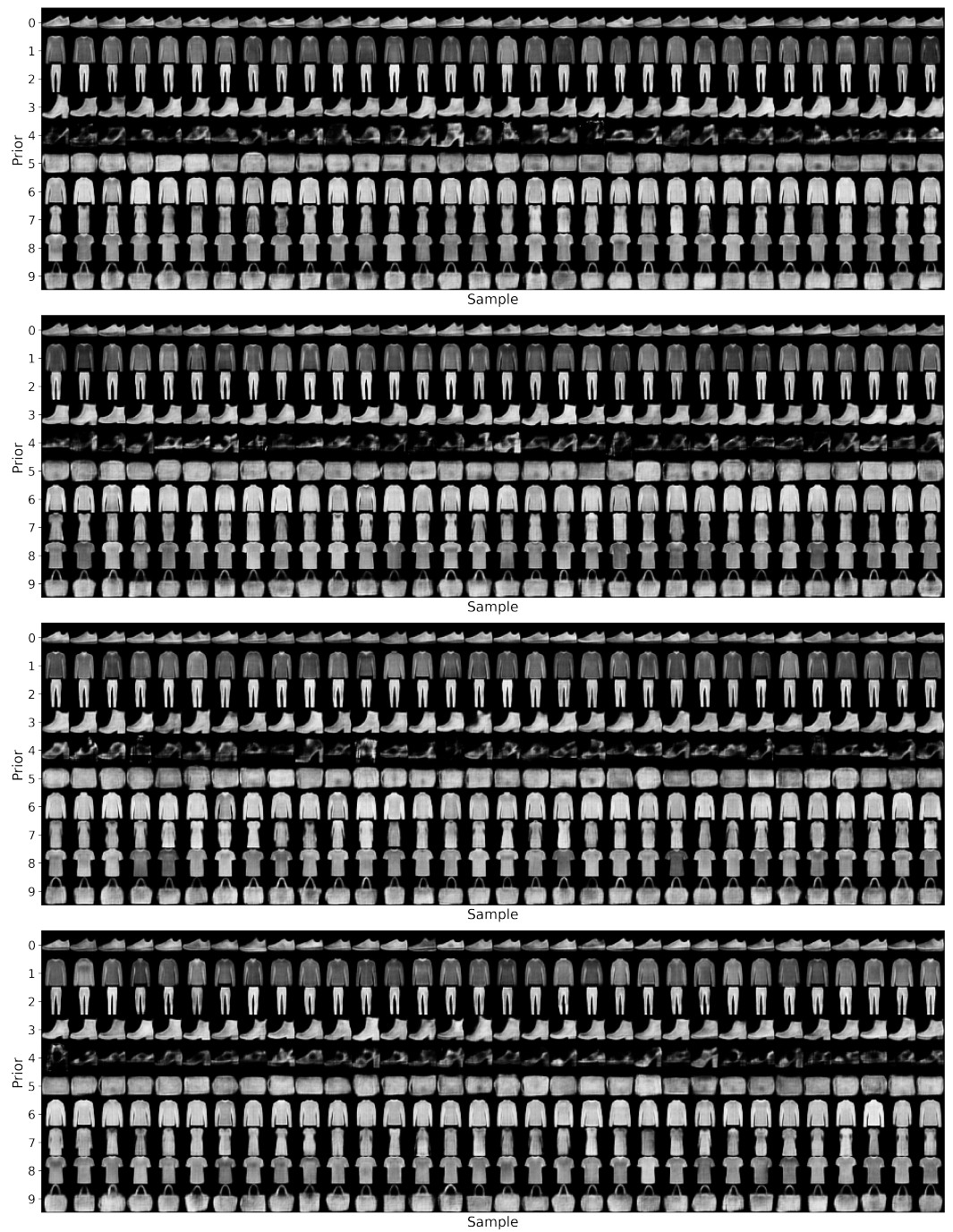

Figure 18: Various samples from each of the generative priors. Each prior learns to represent one of the digits. The DRPM-VC learns nice representations that provide coherent generations of most classes. For high-heels (cluster 4), generating new samples seems difficult due to the heterogeneity within that class.

