# OpenReview forum: "Differentiable Random Partition Models"
_NeurIPS.cc/2023/Conference — NeurIPS 2023 poster_

### Official Review · Reviewer_Ynzj · 2023-07-03

**Soundness:** 4 excellent
**Presentation:** 3 good
**Contribution:** 3 good
**Rating:** 7
**Confidence:** 2

**Summary:**

This paper proposes a probabilistic model for partitioning a non-i.i.d. set of points into a fixed number of subsets, such that the probabilistic model can be fit to the data by variational inference, using the reparameterization trick to enable computation of gradients.  The probabilistic model for partitioning is composed of two processes: one random process, using a multivariate hypergeometric distribution, for determining the sizes of the subsets, and a another random process, which assigns a random ordering to the points according to the Plackett-Luce distribution over permutations, which determines the assignment of points to subsets.  To implement the inference, an evidence lower bound for the marginal probability p(X) is derived.  Three additional methods utilizing the random partitioning model are presented.  One is a method for clustering the data, one is a method for inferring shared and unique latent factors for a pair of data points, and another is a method for partitioning final-layer neurons by task in a multi-task learning setup.

**Strengths:**

The paper shows how random partitioning models have many applications in machine learning, and explains the advantages of this particular approach.  A large number of technical optimizations, such as lower bounds of the loss function, are developed, which seem essential for making the method usable.

**Weaknesses:**

The motivation for the multivariate hypergeometric distribution could benefit from more elaboration.

**Questions:**

* Rather than MVHG, wouldn't it be simpler to use a multinomial distribution instead?
* Under what data generation conditions should the multivariate hypergeometric distribution be advantageous to the multinomial?
* Is the success of MVHG distribution in cases such as MNIST due to the fact that the dataset is constructed to have a fixed number of exemplars per class?  Would we still see an RPM based on MVHG doing better than one based on multinomial distribution in an example where the data has been generated i.i.d.?

**Limitations:**

Yes. The authors mention the memory inefficiency and the possibility of improving the approximations used in the ELBO, which do seem be the most important limitations of the method.

---

> ### Author Rebuttal · Authors · 2023-08-08
>
> Dear reviewer, thank you for your detailed feedback. In the following, we would like to address your questions and concerns.
> >Rather than MVHG, wouldn't it be simpler to use a multinomial distribution instead? Under what data generation conditions should the multivariate hypergeometric distribution be advantageous to the multinomial?\
> >+\
> >Is the success of MVHG distribution in cases such as MNIST due to the fact that the dataset is constructed to have a fixed number of exemplars per class? Would we still see an RPM based on MVHG doing better than one based on multinomial distribution in an example where the data has been generated i.i.d.?
>
>
> We chose the MVHG to model dependencies between elements, which would not be possible using the multinomial distribution. However, the multinomial distribution would also be valid, as the reviewer pointed out. One of the strengths of the proposed two-stage approach is its versatility. Any distribution over subset sizes that is differentiable could be integrated into our approach. Our paper relies on the MVHG distribution due to the mentioned advantages.
> The advantage of using the MVHG distribution will probably vanish if the data is generated truly i.i.d. However, we can rarely guarantee truly i.i.d. data, and the MVHG distribution offers an elegant solution to this problem. In addition, the MVHG implicitly regularizes against collapsing classes, which could happen if using a sampling scheme with replacement.
>
> >The authors mention the memory inefficiency and the possibility of improving the approximations used in the ELBO, which do seem be the most important limitations of the method.
>
> Fortunately, memory inefficiency has not been a problem for our experiments so far, since baseline methods often exhibit similar memory complexity. We provide more details regarding this in our answer to reviewer WEvf. However, for experimental setups where $n$ denotes, e.g., the input dimension, the memory complexity of the DRPM could become an issue and we plan to investigate how to scale our approach to such scenarios in future work.
> Further, to investigate why the approximated ELBO often led to good results in our experiments, we conducted an ablation where we compare the PMF upper/lower bounds to the true PMF for small partitions where the PMF is tractable. We provide a figure showing this ablation in the rebuttal pdf. The results suggest that our approximations lead to a roughly constant offset to the true log PMF. By rescaling the corresponding loss terms with a hyperparameter in our ELBOs, we can thus compensate for the gap induced by our bounds.

---

> > ### Comment · Reviewer_Ynzj · 2023-08-16
> > **Additional questions**
> >
> > Thank you for the answers to my questions.
> >
> > From reading the other reviews and discussions, I realize that you do not cluster the entire dataset in one step, but rather apply the clustering one batch at a time.  I think this fact should be made more apparent in the main paper.  So is $\omega$ the only parameter that is learned across batches?  And do you think it would make sense to include the same point in multiple batches, and then to aggregate the results somehow (similarly to including the same pixel in multiple overlapping windows for image segmentation)?

---

> > > ### Author Response · Authors · 2023-08-17
> > >
> > > Thanks for your reply and the additional questions!
> > >
> > > >From reading the other reviews and discussions, I realize that you do not cluster the entire dataset in one step, but rather apply the clustering one batch at a time. I think this fact should be made more apparent in the main paper.
> > >
> > > Indeed, we cluster the dataset one batch at a time. We briefly mention this at the beginning of Section 5.1 when we make the example with the urn model. However, we acknowledge that this fact may not be entirely apparent and will make sure to add further clarification.
> > >
> > > >So is $\omega$ the only parameter that is learned across batches?
> > >
> > > Generally, we treat the parameters of the prior distributions $p(Y)$ and $p(\mathbf{z})$ as global parameters. We set the corresponding $\mathbf{s}=const.$ and learn $\mathbf{\omega}$, and $\mathbf{\mu}_k$, $\mathbf{\sigma}_k$ for $k\in\\{1, \ldots, K\\}$ globally across batches. On the other hand, we learn all parameters of the approximate posterior distributions $q(\cdot|X)$ as functions of the batch $X$. Finally, we’d like to note that there are no constraints on how the DRPM parameters $\mathbf{s}$ and $\mathbf{\omega}$ should be learned. As demonstrated in the clustering experiment, we can learn them both globally and at batch level. Further, if we want to partition a set on instance-level, we can also learn the parameters per sample as presented in the generative factor partitioning and multitask experiments.
> > >
> > > >And do you think it would make sense to include the same point in multiple batches, and then to aggregate the results somehow (similarly to including the same pixel in multiple overlapping windows for image segmentation)?
> > >
> > > We thank the reviewer for their suggestion! Indeed, it could be interesting to see whether an adaption with overlapping batches could further improve clustering performance in the variational clustering experiment. However, we don’t think such an extension is feasible for the other experiments. As mentioned above, in the other experiments, we do partitioning at instance-level and aggregating batches would thus not influence the predictions.

---

> > > > ### Comment · Reviewer_Ynzj · 2023-08-18
> > > > **Thanks for the clarifications**
> > > >
> > > > I am pleased that you will clarify how batching works in the main paper.  As for my review, I am keeping my score as is.

---

### Official Review · Reviewer_6PXU · 2023-07-06

**Soundness:** 3 good
**Presentation:** 3 good
**Contribution:** 3 good
**Rating:** 7
**Confidence:** 4

**Summary:**

The paper proposes an interesting approach for differentiable partitioning.

**Strengths:**

The writeup of the method is clean and clear and the method is sound.

**Weaknesses:**

While the experimental results demonstrate the strengths of the method, they seem to be pretty tailored for the presented method.
It would be great if there was a more "conventional" experiment where the method can be utilized; this would definitely strengthen the paper, make the approach more convincing, and I would like to offer increasing my score for an experiment that demonstrates the utility in more general settings.

Maybe an experiment from https://arxiv.org/pdf/2305.16358.pdf could also work for your method. If applicable, demonstrating a more direct comparison like this would be very convincing.

As sinkhorn as well as differentiable relaxation of sorting and permutation matrices, it might be good to also include https://arxiv.org/abs/1905.11885 .

**Questions:**

-

**Limitations:**

Limitations are addressed.

---

> ### Author Rebuttal · Authors · 2023-08-08
>
> Dear reviewer, thank you for your detailed feedback. In the following, we would like to address your questions and concerns.
> >While the experimental results demonstrate the strengths of the method, they seem to be pretty tailored for the presented method. It would be great if there was a more "conventional" experiment where the method can be utilized; this would definitely strengthen the paper, make the approach more convincing, and I would like to offer increasing my score for an experiment that demonstrates the utility in more general settings. Maybe an experiment from https://arxiv.org/pdf/2305.16358.pdf could also work for your method. If applicable, demonstrating a more direct comparison like this would be very convincing.
>
> We thank the reviewer for the suggestion. We decided to conduct the “Supervised Clustering” experiment from [5] and present some preliminary results here and in the rebuttal pdf. We train the supervised model by training a small CNN which maps each sample to a representation $\\mathbf{z}$. For each batch, we then use the vectors $\mathbf{z}$ to learn a partition $\\tilde{M}\_\\Omega$ that is trained to align with the actual cluster assignments $M_\\Omega$ similar to [5]. We train the network end-to-end by using the AdamW optimizer and applying the cross-entropy loss between the ground-truth $\tilde{M}\_\\Omega$ and $M\_\\Omega$. In initial experiments, we achieved a precision of 96.47±0.25 on MNIST and 82.71±0.36 on Fashion MNIST without much tuning of network architecture or hyperparameters and running the experiment on 5 seeds each. Similar to [5], we also qualitatively inspected the learned representation by getting the UMAP embeddings in two dimensions. We include these plots in the rebuttal pdf and plan to include our findings in the appendix of the camera-ready version of our manuscript.
>
> We further present additional ablations, such as for our PMF bounds, varying $K$ for clustering, FMNIST generations for $K=20$, clustering experiments on STL-10, and multitasking on CelebA for all $40$ labels in the rebuttal pdf.  We plan to include these in the appendix of the camera-ready version of our manuscript, together with the new supervised clustering ablation, which we compare to [5].
> >As sinkhorn as well as differentiable relaxation of sorting and permutation matrices, it might be good to also include https://arxiv.org/abs/1905.11885 .
>
> We thank the reviewer for the additional reference. We will include it in the camera-ready version of our manuscript.

---

> > ### Comment · Reviewer_6PXU · 2023-08-15
> >
> > Thank you for your detailed response. Accordingly, I raise my score.

---

> > > ### Author Response · Authors · 2023-08-16
> > >
> > > Thank you for your kind response and for raising your score!

---

### Official Review · Reviewer_jtds · 2023-07-07

**Soundness:** 3 good
**Presentation:** 2 fair
**Contribution:** 3 good
**Rating:** 6
**Confidence:** 4

**Summary:**

The authors tackle the problem of partitioning a set of elements into a variable number of disjoint subsets in a differentiable manner. The method starts by first inferring the number of elements per subset. Consequently, they fill these subsets in a learned order.

**Strengths:**

- The problem of partitioning a set of elements into variable number of disjoint sets while remaining amenable to back-prop is indeed an important and timely problem. The authors spend the time to attempt to develop an approach top the problem starting from first principles, which I appreciate.

**Weaknesses:**

- I found the writing to be quite a bit confusing and very notation heavy at times, with very little by way of intuition or meaning of the equations. A running example, e.g. an urn with marbles, would've greatly helped with the exposition.

- After reading the paper, I'm still not really sure how the authors overcome the non-differentiability of sampling.

- I'm not sure I get the point behind Lemma 4.3? It doesn't add much, and doesn't really mention how differentiability is achieved (I'm guessing through Gumbel-Softmax?)

- The related work can be further improved as it's missing a lot of work regarding differentiable discrete distributions e.g. [1], [2], [3]

References:

[1] Marin Vlastelica, P., Anselm Paulus, Vít Musil, Georg Martius and Michal Rolinek. “Differentiation of Blackbox Combinatorial Solvers.” ArXiv abs/1912.02175 (2019)

[2] Mathias Niepert, Pasquale Minervini and Luca Franceschi. “Implicit MLE: Backpropagating Through Discrete Exponential Family Distributions.” Neural Information Processing Systems (2021).

[3] Kareem Ahmed, Zhe Zeng, Mathias Niepert and Guy Van den Broeck. “SIMPLE: A Gradient Estimator for k-Subset Sampling.” ICLR 2023.

**Questions:**

- In equation 2, what purpose does the superscript in $\omega_k^{n_k}$ serve? It is my understanding that the weights are specific to each class, not each instance? Also, am I correct in my understanding that $P_0$ here is intractable?

- I'm a bit confused as to how equation (2) induces a distribution over subset sizes (as per the paragraph title). I see that induces a distribution over the assignments to classes e.g. (2, 3, 0) would denote that we sampled 2 of class 0, 3 of class 1 and 0 of class 1. That should be enough to induce a distribution over subset sizes only if you assume somehow that each subset is comprised of items of a single class? What do we lose here, if anything, by making such an assumption?

- If I'm not mistaken, Equation (3) is very similar to weighted reservoir sampling (see e.g. [1])?

- It is not clear how you overcome the non-differentiability of sampling. Do you use Gumbel-Softmax when sampling both the permutation
matrix as well as an order?

- Am I correct in my understanding that equation (5) is simply stating that the probability of subsets $\mathbf{y}_1, \ldots, \mathbf{y}_n$ is simply the probability of the subsets sizes $\mathbf{n}$ multiplied by the probability of all permutations of the elements of in each subset $\mathbf{y}_i$?


References:

[1] Sang Michael Xie and Stefano Ermon. “Reparameterizable Subset Sampling via Continuous Relaxations.” International Joint Conference on Artificial Intelligence (2019).

**Limitations:**

Adequately addressed.

---

> ### Author Rebuttal · Authors · 2023-08-08
>
> Dear reviewer, thank you for your detailed feedback. In the following, we would like to address your questions and concerns.
> >I found the writing to be quite a bit confusing and very notation heavy at times, with very little by way of intuition or meaning of the equations. A running example, e.g. an urn with marbles, would've greatly helped with the exposition.
>
> We like the proposition of including a marble example and propose integrating the following paragraph into the camera-ready version of our manuscript:
>
> Within the scope of our work, a partition of a set of $n$ elements can be viewed as a special case of the urn model. Here, the urn contains marbles with $n$ different colors, where each color corresponds to a subset in the partition. For each color, there are $n$ marbles corresponding to the potential elements of their color/subset. To derive a partition, we sample $n$ marbles without replacement from the urn and register the order in which we draw the colors. The color of the $i$th marble then determines the subset to which element $i$ corresponds. Furthermore, we can constrain the partition to only $K$ subsets by taking an urn with only $K$ different colors.
>
> We thank the reviewer for suggesting the idea of a running example. We would further like to note that we attempted to make the notation as concise and consistent as possible. However, it lies within the nature of the problem that a lot of notation is needed to adequately describe our work. We would be happy to take propositions regarding cleaner notation into consideration for the final manuscript.
> > After reading the paper, I'm still not really sure how the authors overcome the non-differentiability of sampling.\
> > +\
> > I'm not sure I get the point behind Lemma 4.3?[...] \
> > +\
> > It is not clear how you overcome the non-differentiability [...]
>
> We outline the technical details to achieve differentiability in the proof of Lemma 4.3 in the appendix. In short, we apply variations of the Gumbel-Softmax trick and straight-through estimators [3] to get hard samples of the order/permutation $\pi$ and elements per subset $\mathbf{n}$ in the forward pass while retaining soft gradients in the backward pass. Given $\\nu_k:=\\sum\_{i=1}^{k-1}\\mathbf{n}\_i $
> and $\\nu\_{k+1}$, we compute a vector $\\mathbf{\\tilde{\\alpha}}\_k\\in[0,1]^n$ where $\\mathbf{\\tilde{\\alpha}}\_k^{(i)}=1$ if $i\in[\nu_k,\nu_{k+1}]$ and otherwise $\\mathbf{\\tilde{\\alpha}}\_k^{(i)}=0$. We then use $\\mathbf{\\tilde{\\alpha}}$ to remove all entries from $\pi$ that do not correspond to subset $k$ by multiplying it columnwise and then retrieve $\mathbf{y}_k$ by adding up the resulting matrix row-wise, similar to the example in Figure 1.
> >The related work can be further improved [...]
>
> We thank the reviewer for pointing out the additional references. We agree that they are related to our work and will add them to the camera-ready version of our manuscript.
> >In equation 2, what purpose does the superscript in $\omega_k^{n_k}$ serve? It is my understanding that the weights are specific to each class, not each instance? Also, am I correct in my understanding that $P_0$ here is intractable?
>
> Yes, the weights $\omega_k^{n_k}$ are class/subset weights and not instance weights. The score $s_i$ for element $i$ could be interpreted as an instance weight. The superscript $n_k$ connects the number of drawn elements per class $n_k$ to the class importance $\omega_k$ of every class $k$.
> Regarding the MVHG distribution, we follow the sequential procedure introduced in [4] instead of directly applying eq. (2). In eq. (2), $P_0$ is the distribution-wide normalization parameter. Using class-conditional distributions, the number of states per conditional distribution reduces to $n$, allowing tractable normalization. However, you are correct that $P_0$ is intractable if we directly apply eq. (2). For more details, we refer to [4].
> >I'm a bit confused as to how equation (2) induces a distribution over subset sizes [...]
>
> A partition model is defined as assigning every element $i$ to one of $K$ subsets. The content of each subset itself depends on the specific task we want to solve. In the proposed two-stage approach, we use the MVHG as a distribution over subset sizes and the PL distribution to model the content of each subset. Going back to the initial marble example, the MVHG models the number of marbles we draw per color, whereas the PL distribution models the order in which we draw the marbles. In this setting, a subset can be defined arbitrarily, and the MVHG still defines a distribution over subset sizes. The distribution over subset sizes does not make any assumptions about the content of each subset.
> In our clustering experiment, the reviewer's observation is correct that we see the different classes of MNIST/FMNIST as subsets, whereas in the other experiments, subsets denote groups of shared/independent generative factors or neurons specialized to a single task.
> We hope this addresses your question and are happy to answer any further questions regarding Equation (2).
> > If I'm not mistaken, Equation (3) is very similar to weighted reservoir sampling (see e.g. [1])?
>
> Equation (3) is indeed related to [1], as both originate from [2], where they show the relationship between Luce’s choice axiom and the Gumbel-max trick. We list [1] in our related work and reference [2] in the preliminaries.
> > Am I correct in my understanding that equation (5) is simply stating that the probability of subsets $\mathbf{y}_1, \ldots, \mathbf{y}_n$ is [...]
>
> The reviewers' observation is indeed correct and stems from our two-stage procedure. For each partition $Y$, there exists a single unique $\mathbf{n}$ that leads to that partition. However, all permutations in $\Pi_Y$ can also lead to $Y$, which is why we need to sum over their probabilities. For a more intuitive example and more details, we refer to Figure 1 and the proof of Proposition 4.1 in Appendix B.

---

> > ### Comment · Reviewer_jtds · 2023-08-15
> >
> > Thanks for you response.
> >
> > [Writing Clarity]
> >
> > I appreciate the technical nature of the topic, requiring sufficient notation to describe the problem. I do not, however, believe that contradicts with the clarity of exposition. For instance, equation (2) is In my opinion, stated without any commentary beyond it corresponding to the probability of sampling $\mathbf{n}$ and $P_0$ being the normalization constant. Another instance of this is your re-statement of the problem as an urn model. Unless I'm mistaken, I do not see the need to denote by $n$ both the number of different colors in the urn as well as the number of marbles per color and then *again* the sample size?
> >
> > [Differentiability]
> >
> > As a paper on *differentiable* subset sampling I feel like perhaps the means by which you render such an operation differentiable should come earlier, with more discussion, instead of simply being relegated to the appendix.
> >
> > Other than that, I believe you've answered my concerns. I'm raising my score to a weak accept.

---

> > > ### Author Response · Authors · 2023-08-15
> > > **Official comment by the authors**
> > >
> > > Thanks for your reply!
> > >
> > > We would like to respond to your comments and clarify them.
> > >
> > > > For instance, equation (2) is In my opinion, stated without any commentary beyond it corresponding to the probability of sampling $\mathbf{n}$ and $P_0$ being the normalization constant.
> > >
> > > We agree that we keep the description of equation (2) short. However, due to the page limit, we decided to keep the preliminaries relatively compact and refer to the appendix for more details.
> > >
> > >
> > > > Unless I'm mistaken, I do not see the need to denote by n both the number of different colors in the urn as well as the number of marbles per color and then again the sample size?
> > >
> > > Thanks for your question. We want to clarify that we denote the number of different colors, the number of marbles per color, and the sample size by $n$ because in our version of the urn model (partition model) they all have the same value.
> > > What often leads to confusion is the number of subsets. Note that, at most, there can be as many subsets as there are elements in the set we want to partition, namely if each element is placed in a separate subset. Since each color stands for one subset, there are a maximum of $n$ colors. Our paper follows previous work [a], which introduced $K$ as the number of subsets where $K \leq n$.
> > > Further, in our preliminaries section of the MVHG distribution and equation (2), we initially follow the notation of [4] and state that there are $m_k$ marbles of color $k$ in the urn. In the case of a partition, each subset can contain, at most, as many elements as there are elements in the set we want to partition. Thus, we set the number of marbles for each color to $n$, i.e., $\forall k\in[K]: m_k=n$.
> > > Hence, we decided to keep $n$ as a central part of our notation instead of introducing additional notation that denotes the same value.
> > >
> > > [a] Mansour, T., & Schork, M. (2015). Commutation relations, normal ordering, and Stirling numbers. CRC Press.
> > >
> > >
> > > > As a paper on differentiable subset sampling I feel like perhaps the means by which you render such an operation differentiable should come earlier, with more discussion, instead of simply being relegated to the appendix.
> > >
> > > We decided to focus the main part of our paper on the idea of the two-stage procedure, which we view as our main contribution, and keep the technical details brief. In our opinion, this improves the clarity of our paper and helps in understanding the core part of our contribution. However, we see your point, and without space constraints, we would have added more discussion on the differentiability aspect of the proposed formulation.

---

### Official Review · Reviewer_WEvf · 2023-07-09

**Soundness:** 3 good
**Presentation:** 3 good
**Contribution:** 3 good
**Rating:** 6
**Confidence:** 3

**Summary:**

This paper proposes a two-stage differentiable partitioning method where the first stage outputs number of elements in each cluster and  the second stage assigns samples by ordering them through a permutation matrix. The differentiability in their formulation is achieved via gumbel-softmax trick for distributions sampling. The paper shows the performance of their method for 1-a variational clustering task on MNIST nad FMNIST, 2-learn a partition of generative factors on mpi3d dataset, 3-neuron grouping task for multi-task learning on MultiMNIST.

**Strengths:**

- The proposed formulation and its usage in a differentiable setting is novel from my knowledge. Further applying this method for grouping neural networks for MTL and other tasks are novel from my knowledge. However, I am not an expert in this field, so I may not be aware of some of the previous work.

- The paper is clearly written with a decent language and flow. Supplementary material is also very comprehensive and helpful for people who are not very familiar to the field (although I did not check all the details). For example, it introduces random partitioning well, carefully states the definitions and set up. Experiment set up and results are provided in detail. One suggestion is some of the material related to the method could be carried from supplementary to the main paper, such as final loss function, model plot, two stage algorithm details.

- Although the applicability to a less trivial setting -- with larger datasets and harder tasks -- are not clear, the method shows promise to be improved and has sufficient contribution. Sampling from cluster numbers and permutation matrix distributions in a differentiable way to create partitioning is an interesting idea and application.

**Weaknesses:**

- The experiment results need more clarification for a couple of concerns I have to understand the impact and contribution better:

    - Why some of the recent differentiable clustering papers are missing in terms of both citation and baselines(examples: [1,2,3])? Is it because only generative clustering is considered? If that is the reason, it needs to be mentioned somewhere in the paper, perhaps either in related work or introduction section.

    - Why Table 1 especially ACC results are very different (low) than how the baselines' original papers reported (for example VADE paper)? Is the network used different, or set up? It needs to be clarified, and maybe another table with the same set up can be included.

    - Why didn't you try any dataset other than MNIST for clustering tasks, for instance STL-10 or Reuters datasets at least?

- If your main argument relies on contributing to the generative clustering methods, again introduction may be modified to reflect that. Otherwise, the paper needs to include other differentiable clustering/partitioning methods such as deep k-means and [1,2,3].








1) Yang, Xu, et al. "Deep spectral clustering using dual autoencoder network." Proceedings of the IEEE/CVF conference on computer vision and pattern recognition. 2019.
2) Cai, Jinyu, et al. "Efficient deep embedded subspace clustering." Proceedings of the IEEE/CVF Conference on Computer Vision and Pattern Recognition. 2022.
3) Sarfraz, Saquib, Vivek Sharma, and Rainer Stiefelhagen. "Efficient parameter-free clustering using first neighbor relations." Proceedings of the IEEE/CVF conference on computer vision and pattern recognition. 2019.

**Questions:**

- A quadratic relationship wrt n is mentioned in the limitations. How does your method's computational complexity compare with the baselines? Is this why you did not go beyond MNIST?

- Why you experimented with fixed encoder and different initialization? (in the supplementary)



**Limitations:**

Yes

---

> ### Author Rebuttal · Authors · 2023-08-08
>
> Dear reviewer, thank you for your detailed feedback. In the following, we would like to address your questions and concerns.
>
> >Why some of the recent differentiable clustering papers are missing in terms of both citation and baselines(examples: [1,2,3])? Is it because only generative clustering is considered? If that is the reason, it needs to be mentioned somewhere in the paper, perhaps either in related work or introduction section.\
> >+\
> >If your main argument relies on contributing to the generative clustering methods, [...]
>
> Thank you for pointing this out. Our first experiment focuses on comparing generative clustering methods. Generative clustering approaches offer the unique advantage of enabling unsupervised conditional generation, with the tradeoff of having slightly worse clustering performance as opposed to more conventional methods. Consequently, a direct comparison with non-generative clustering methods might not be entirely fair. However, we would like to note that our DRPM offers the opportunity to develop other types of clustering algorithms, such as tree- or contrastive learning-based approaches if generative modeling is not desired. We will address this point in the camera-ready version and we will add the mentioned papers to our related work section.
> >Why Table 1 especially ACC results are very different (low) than how the baselines' original papers reported (for example VADE paper)? Is the network used different, or set up? It needs to be clarified, and maybe another table with the same set up can be included.
>
> For the VADE experiments, we maintained consistency with the original VADE paper by using the same optimizer, architecture, and loss function. However, we would like to point out two discrepancies between our setup and what was reported in the original paper. First, the original VADE experiments did not have a test set but were trained and evaluated on the full dataset (train+test split) as can be seen from their code (https://github.com/slim1017/VaDE). Furthermore, in contrast to reporting average statistics, the original paper initialized the method ten times and then reported the performance of the model with the best objective value, as described in Section 4.3 of [6]. Both these aspects do not align with current best practices, and we hence used separate train/test splits in our experiments and decided to report average statistics, including standard deviations over several seeds. Finally, we added an ablation where we tuned the VADE baseline by employing certain optimizations that proved successful in the case of our DRPM-VC model. Using these optimizations, we were able to boost the average performance of VADE close to what was reported in the original paper, as shown in Table 5 of the appendix.
> >Why didn't you try any dataset other than MNIST for clustering tasks, for instance STL-10 or Reuters datasets at least?
>
> The goal of our paper is to introduce the DRPM and demonstrate its versatility in a number of different scenarios. The focus of our experiments does not solely lie in raw performance on numerous datasets but also on demonstrating how the DRPM can be integrated into a variety of different scenarios as a handy tool that helps to incorporate new generative assumptions (Variational Clustering/Generative Factor Partitioning) or circumvent costly hyperparameter tuning (Multitask experiment).
> However, to demonstrate that we can easily extend our approach to other datasets, we conducted experiments on the STL-10 dataset following the setup of [6] for this rebuttal (see rebuttal pdf).
> >A quadratic relationship wrt n is mentioned in the limitations. How does your method's computational complexity compare with the baselines? Is this why you did not go beyond  MNIST?
>
> We would like to note that for the experiments “Variational Partitioning of Generative Factors” and “Multitask Learning”, including DRPM does not increase memory complexity compared to baselines. Here, $n$ corresponds to the number of latent dimensions and number of neurons respectively. Since both model architectures employ fully connected layers to derive the respective intermediate representation, memory complexity is $\Omega(n^2)$ for all baselines. In contrast, for the variational clustering experiment, additional memory complexity is added since $n$ denotes the batch size. However, $n$ is thus independent of the dataset and not a limiting factor since we can always choose it to be smaller if necessary.
>
> In practice, we never encountered issues where we had to compromise on the value of $n$. For instance, we ran the variational clustering experiments with a batch size of $n=256$, partitioned the latent space $n=10$ on the mpi3d dataset, and added multitask experiments on CelebA with $n=64$ to the appendix (Section C.3). Note that all our experiments were conducted on an RTX2080Ti GPU. For this rebuttal, we also added an additional ablation for CelebA where we partitioned the neurons for all $40$ tasks without running into memory issues. These experiments highlight the versatility of the proposed method and that our approach is generally scalable to more difficult datasets.
> >Why you experimented with fixed encoder and different initialization? (in the supplementary)
>
> Similar to the VADE paper, we realized that proper initialization of the cluster parameters and network weights is crucial for the success of variational clustering. We initially worked with the originally proposed autoencoder pretraining (Appendix C.2.4.) but found that incorporating more recent advances from the contrastive literature improves the clustering performance of both VADE and DRPM-VC. Further, we found that fixing early encoder layers leads to more stable training and better results (Appendix C.2.5). However, independent of the strategy used, we consistently outperform our baselines, suggesting that the DRPM may be a better prior model than i.i.d. Categorical for variational clustering.

---

> > ### Comment · Area_Chair_GARg · 2023-08-19
> > **Thanks for the response**
> >
> > Dear authors,
> >
> > Thanks for your efforts to clarify the concerns raised by the reviewer. Although the reviewer has not responded yet, I'd like to assure you that your response will be incorporated into the reviewers' discussion and my final recommendation.
> >
> > Best wishes,
> > AC

---

### Author Rebuttal · Authors · 2023-08-08

We would like to thank all reviewers for providing comprehensive and valuable feedback. We particularly value the reviewers' recognition of the novelty (WEvf), importance (jtds), soundness (6PXU), and applicability (Ynzj) of our work. Your insightful comments and suggestions have been immensely beneficial in enhancing the overall quality of our paper.

Your thorough reviews have helped us identify potential areas for improvement and gain valuable perspectives that undoubtedly strengthen the contribution of our research. We have carefully addressed each of your comments and will incorporate your feedback into the camera-ready version of our paper. We believe that your inputs will contribute to the advancement of our work.

We welcome any additional suggestions, questions, or requests for information and encourage further discussions with all reviewers. We hope that our efforts in adequately addressing the reviewers' questions or concerns will lead them to consider adjusting their scores or confidences.

Once again, we thank the reviewers for their time and effort in evaluating our work.


**Rebuttal References:**

[1] Sang Michael Xie and Stefano Ermon. “Reparameterizable Subset Sampling via Continuous Relaxations.” International Joint Conference on Artificial Intelligence (2019).

[2] J. I. Yellott. The relationship between Luce’s choice axiom, Thurstone’s theory of comparative judgment, and the double exponential distribution. Journal of Mathematical Psychology, 15(2): 109–144, 1977. Publisher: Elsevier.

[3] Y. Bengio, N. Léonard, and A. Courville. Estimating or propagating gradients through stochastic neurons for conditional computation. arXiv preprint arXiv:1308.3432, 2013.

[4] Sutter, Thomas M., et al. "Learning Group Importance using the Differentiable Hypergeometric Distribution." arXiv preprint arXiv:2203.01629 (2022).

[5] Stewart, Lawrence, et al. "Differentiable Clustering with Perturbed Spanning Forests." arXiv preprint arXiv:2305.16358 (2023).

[6] Jiang, Zhuxi, et al. "Variational deep embedding: An unsupervised and generative approach to clustering." arXiv preprint arXiv:1611.05148 (2016).

[7] Marin Vlastelica, P., Anselm Paulus, Vít Musil, Georg Martius and Michal Rolinek. “Differentiation of Blackbox Combinatorial Solvers.” ArXiv abs/1912.02175 (2019)

[8] Mathias Niepert, Pasquale Minervini and Luca Franceschi. “Implicit MLE: Backpropagating Through Discrete Exponential Family Distributions.” Neural Information Processing Systems (2021).

[9] Kareem Ahmed, Zhe Zeng, Mathias Niepert and Guy Van den Broeck. “SIMPLE: A Gradient Estimator for k-Subset Sampling.” ICLR 2023.

---

### Decision · Program_Chairs · 2023-09-21

**Decision:**

Accept (poster)

**Comment:**

This paper presents a novel and significant contribution, where a "differentiable" procedure of partitioning a given set of items into a variable number of blocks is presented. The presented algorithm would be useful for application in many areas of machine learning. The reviewers are generally happy with the novelty of the proposed method and the quality of the empirical validation.